# Tropopause altitude determination from temperature profile measurements of reduced vertical resolution

Nils König[1], Peter Braesicke[1], and Thomas von Clarmann[1]

[1]Karlsruhe Institute of Technology, Institute of Meteorology and Climate Research,
Karlsruhe, Germany

**Correspondence:** T. von Clarmann (thomas.clarmann@kit.edu)

**Abstract.** Inference of the lapse rate tropopause or the cold point from temperature profiles of finite vertical resolution entails an uncertainty of the tropopause altitude. For tropical radiosonde profiles the tropopause altitude inferred from coarse grid profiles was found to be lower than that inferred from the original profiles. The mean (median) displacements of the lapse rate tropopause altitude when inferred from a temperature profile of 3 km vertical resolution and a Gaussian kernel are -130 m, -400 m, -730 m, and -590m (-70 m, -230 m, -390 m, and -280 m) for Nairobi, Hilo, Munich, and Greifswald, respectively. In case of a Michelson Interferometer for Passive Atmospheric Sounding (MIPAS) averaging kernel the displacement of the lapse rate tropopause altitude is -640 m. The mean (median) displacement of the cold point tropopause inferred from a temperature profile of 3 km vertical resolution (Gaussian kernels) was found to be -510 m, -610 m, -530 m, and -390 m (-460 m, -510 m, -370 m, and -280 m) for the stations mentioned above. Unsurprisingly, the tropopause altitude displacement is larger for coarser resolutions. The effect of the tropopause displacement on the water vapour saturation mixing ratio is roughly proportional to the vertical resolution. In tropical latitudes the resulting error is about one to two parts per million by volume per vertical resolution in km. The spread of the tropopause displacements within each sample of profiles seems too large as to recommend a correction scheme for tropical temperature profiles, while for midlatitudinal temperature profiles of vertical resolutions of 1 to 5 km a lapse rate of -1.3 K/km reproduces tropopause altitudes determined from high-resolution temperature profiles with the nominal lapse rate criterion of -2 K/km fairly well.

## 1 Introduction

The tropopause constitutes a vertical separation in the atmosphere that segregates the lower, weather active region, *viz.*, the troposphere, from an upper, steadier region, the stratosphere. High altitude temperature soundings that became possible at the end of the 19th century showed an – at that time – unexpected temperature behaviour, where temperatures would stagnate or even increase with height (see Hoinka, 1997, for a historical overview). Once it was established that this observation was no measurement error, and that above the troposphere another region of the atmosphere exists, namely the stratosphere, an unambiguous definition for the height of the boundary, the tropopause, had to be agreed. The earliest comprehensive definition provided by the British Meteorological Office was based on either the existence of a temperature inversion, or an abrupt transition to a temperature gradient below 2 K/km. If the first two criteria were not met, a more general vertical temperature

gradient criterion was applied: "at the point where the mean fall of temperature for the kilometre next above is 2 K or less provided that it does not exceed 2 K for any subsequent kilometre" Dines (1919, cited after Hoinka, 1997). A similar definition, focusing solely on the lapse rate of 2 K/km was adapted by the World Meteorological Organization (WMO) in later years (World Meteorological Organization, 1957). Since then additional definitions of the tropopause have emerged, focusing on the

behaviour of dynamical quantities (e.g. Hoerling et al., 1991) or of trace gas changes (e.g. Pan et al., 2004). However, the most commonly used method to define the position of the tropopause still is the WMO criterion.

In tropical latitudes, another useful reference for distinguishing the tropopause from the stratosphere is the cold point (where the temperature minimum occurs). It relates to the existence of a temperature inversion in the original definition as described above and the corresponding lapse rate tropopause lies commonly a few hundred meters below the cold point (e.g., Figure 8 in

Kim and Son, 2012).

Aspirational targets exist for knowing the altitude distribution of the thermal tropopause with an uncertainty of 100 m globally (see the Observing Systems Capability Analysis and Review Tool at https://www.wmo-sat.info/oscar/variables/view/81). However, it is obvious that deriving the altitude of a lapse rate tropopause will depend to some extent on the resolution of the temperature profile that is used to calculate the vertical gradient. The same holds true for the cold point tropopause. Thus, it

seems important to understand how the derived altitude of the tropopause depends on the vertical resolution of the temperature data. Knowledge of the exact tropopause altitude is essential in particular, when distributions of atmospheric state variables such as mixing ratios of trace species are transformed to a tropopause-related vertical coordinate system in order to investigate chemical, transport and mixing processes in the upper troposphere and lowermost stratosphere (e.g. Tuck et al., 1997; Pan et al., 2004; Birner, 2006; Tilmes et al., 2010; Pan and Munchak, 2011). Tropopause altitudes inferred from limb measurements have

been reported by, e.g., Peevey et al. (2012) or Spang et al. (2015).

The goal of this paper is to analyze the possible dependence of a derived tropopause altitude on the vertical resolution of the temperature profile and to evaluate possibilities to potentially correct tropopause altitudes inferred from coarsly resolved temperature profiles. After presenting the formal concept used for this study (Section 2), we first assess the impact of finite vertical resolution on the determination of the tropopause altitude in quantitative terms (Section 3). We do this separately for

lapse rate tropopause altitudes (Section 3.1) and cold point tropopause altitudes (Section 3.2). The implication for the saturation mixing ratio of water vapour is analyzed in Section 4. Then we investigate if related altitude errors can be corrected by a slight modification of the tropopause definition which, when applied to temperature profiles of finite vertical resolution, reproduces the tropopause altitude according to the WMO definition when applied to the original data (Section 5). Finally we discuss the applicability of our results to various types of constrained temperature retrievals from satellite data and conclude what the

upshot of this study is from a data user perspective (Section 6).

## 2   The formal concept

The altitude resolution of a vertical profile such as temperature can be characterized by the $n \times n$ averaging kernel matrix **A** (Rodgers, 2000). It consists of the partial derivatives $\frac{\partial \tilde{x}_i}{\partial x_j}$ of the elements $\tilde{x}_i$ of the degraded profile with respect to the variation

of the element $x_j$ of the true profile. Its columns represent the relative response of a degraded profile $\tilde{x}$ to a delta perturbation of the true profile $x$. Conversely, the $j$th column represents the weights with which the elements of the true profile contribute to the $j$th element of the degraded profile. The averaging kernel of a profile without degradation is the identity matrix $\mathbf{I}$. It goes without saying that effects on a finer scale than that reproducible in the $n$-dimensional grid remain undetected, unless

some prior information on the profile shape between the gridpoints is used, as suggested by, e.g., Reichler et al. (2003). This is to say, the averaging kernel does not characterize the degradation with respect to the fully resolved true profile but only the degradation with respect to the profile represented in a vector of $n$ gridpoints.

Typically, the reduction of altitude resolution is caused by one of the following three mechanisms: (1) the atmosphere is remotely sensed by an instrument of finite vertical resolution. In this case, the atmospheric state is often sampled on a grid finer

than that corresponding to the altitude resolution of the measurement system; (2) A high-resolution profile is resampled on a coarser grid. This resampling goes along with a degradation of the altitude resolution; (3) a filter function is applied, which reduces the vertical resolution.

## 2.1  Remotely sensed vertical profiles

Often the degradation, i.e., the loss of vertical resolution, is caused by the use of a constraint in the retrieval of atmospheric

state variables from remote measurements $y$. The estimated state $\hat{x}$ depends on the measurement $y$ and the prior information $x_\mathrm{a}$ as

$$\hat{x} = x_\mathrm{a} + \left(\mathbf{K}^T \mathbf{S}_\mathrm{y}^{-1} \mathbf{K} + \mathbf{R}\right)^{-1} \mathbf{K}^T \mathbf{S}_\mathrm{y}^{-1} \left(y - f(x_\mathrm{a})\right) \tag{1}$$

where $\mathbf{K}$ is the Jacobian matrix $\frac{\partial y_i}{\partial x_j}$, $^T$ indicates a transposed matrix, $\mathbf{S}_\mathrm{y}$ is the covariance matrix characterizing measurement noise, $\mathbf{R}$ is a regularization matrix, and $f$ is the radiative transfer function (von Clarmann et al., 2003a). Using an inverse a

priori covariance matrix $\mathbf{S}_\mathrm{a}^{-1}$ as regularization matrix, this formalism renders a maximum a posteriori retrieval as described by Rodgers (2000). Other widely used choices of $\mathbf{R}$ are squared $l$th order difference matrices (see, e.g. von Clarmann et al., 2003b). The latter are often used in order to stabilize the profile by smoothing without pushing the values towards an a priori profile (e.g. Steck and von Clarmann, 2001).

In all cases, the averaging kernel matrix is

$$\mathbf{A}_\mathrm{retrieval} = \left(\mathbf{K}^T \mathbf{S}_\mathrm{y}^{-1} \mathbf{K} + \mathbf{R}\right)^{-1} \mathbf{K}^T \mathbf{S}_\mathrm{y}^{-1} \mathbf{K}, \tag{2}$$

and with this the state estimate can be separated into two components, which are the contribution of the true atmospheric state and the contribution of the prior information

$$\begin{aligned} \hat{x} \;=\; & \mathbf{A}_\mathrm{retrieval} x_\mathrm{true} \\ & + (\mathbf{I} - \mathbf{A}_\mathrm{retrieval}) x_\mathrm{a} + \epsilon_{x;total} \end{aligned} \tag{3}$$

where, as its index suggests, $x_\mathrm{true}$ represents the true temperature profile, and $\epsilon_{x;total}$ is the actual realization of the retrieval error.

The altitude resolution of the retrieval can be determined from the averaging kernel matrix. Common conventions are to either use the halfwidths of its rows or the gridwidths divided by the diagonal elements. It goes without saying that the altitude resolution of a retrievend profile can be altitude-dependent.

## 2.2 Resampling on a coarser grid

Other causes for degraded profiles are representation on a grid not sufficiently fine to represent all structures or application of a numerical filter to the original profile. The averaging kernel matrix is the adequate tool for dealing with all these cases.

The effect of a coarse grid is best understood by construing the coarse-grid profile as a result of an interpolation of the profile from a finer grid (see Rodgers 2000, Sect 10.3.1, where a slightly different notation is used). Let $\tilde{x}$ be the profile represented on the coarse grid, and $x$ the profile in the original representation where all fine structure is resolved. In this case we use an interpolation matrix $\mathbf{V}$ and get

$$\tilde{x} = \mathbf{V}x \tag{4}$$

For an interpolation from a fine grid to a coarse grid $\mathbf{V}$ is often chosen as

$$\mathbf{V} = (\mathbf{W}^T\mathbf{W})^{-1}\mathbf{W}^T, \tag{5}$$

where $\mathbf{W}$ is the interpolation from the coarse to the fine grid. A definition of $\mathbf{A}$ based on $\mathbf{V}$ gives us an asymmetric averaging kernel matrix which represents the dependence of the profile values on the coarse grid on the "true" values on the fine grid.

$$\mathbf{A}_{\mathrm{coarse}} = \mathbf{V} \tag{6}$$

Contrary to the averaging kernel matrix introduced by Eq (2), $\mathbf{A}_{\mathrm{coarse}}$ is not quadratic.

To characterize the loss of resolution due to coarse sampling the averaging kernel on the fine grid, $\mathbf{A}_{\mathrm{interpolation}}$, is needed. It is

$$\mathbf{A}_{\mathrm{interpolation}} = \mathbf{W}\mathbf{V}. \tag{7}$$

If the profile on the fine grid is in itself a degraded profile, e.g., because it was generated by a constrained retrieval, we need a combined averaging kernel matrix

$$\mathbf{A}_{\mathrm{combined}} = \mathbf{W}\mathbf{V}\mathbf{A}_{\mathrm{retrieval}}. \tag{8}$$

## 2.3 Application of filter functions

Application of a linear filter corresponds to the convolution of the original profile with a filter function and is best formulated as a matrix product involving a filter matrix $\mathbf{T}$ whose lines correspond to the moving discretized filter functions at its actual position.

$$\tilde{x} = \mathbf{T}x \tag{9}$$

In this case the averaging kernel matrix is identical to the $\mathbf{T}$ matrix.

$$\mathbf{A}_{\text{filter}} = \mathbf{T} \tag{10}$$

## 3 The dependence of the estimated tropopause altitude on vertical resolution of the underlying temperature profile

To analyze the impact of smoothing effects on the estimated tropopause altitude we use temperature profiles measured by ra-
diosondes launched from Nairobi (1.3°S, 36.8°E), Hilo (19.4°N 155.4°W), Munich (47.8°N,10.9°E), and Greifswald (54.1°N,
13.4°E). Data from Nairobi and Hilo were available via the Southern Hemisphere Additional Ozonesondes (SHADOZ) net-
work (https://tropo.gsfc.nasa.gov/shadoz/, retrieved on 20 May 2017 Witte et al. 2017; Thompson et al. 2017), while data from
Munich and Greifswald were obtained from the German Weather Service (available via ftp://ftp-cdc.dwd.de/). All radiosonde
data sets cover the period 2007-2018. All available Nairobi and Hilo radiosonde profiles within this time period were used.
For Munich and Greifswald one profile per week was selected. Details of the sonde profiles used in our study are compiled in
Table 1. One focus of our study lies on tropical temperature profiles, because of the importance of the tropical tropopause in
the climate system. Obviously, the area of the tropics exceeds that of other latitude bands; the tropical tropopause is the entry
point of air into the stratosphere (Fueglistaler et al., 2009); and finally the tropical tropopause region plays a distinctive role
in the radiative budget of the Earth (Riese et al., 2012). Nairobi was chosen as an example of a continental station, while Hilo
(Hawaii) is a maritime station. As a contrast we have also used the two midlatitudinal stations Munich (close to the Alps) and
Greifswald (close to the Baltic Sea).

Various averaging kernels $\mathbf{A}$ are applied to the original radiosonde profiles $\mathbf{x}$ to get degraded temperature profiles $\tilde{x}$. These
are used for the determination of the lapse rate tropopause altitude and the results are then compared to the tropopause altitudes
determined from the original sonde data.

### 3.1 The Lapse Rate Tropopause

The lapse rate tropopause is the lower boundary of the lowermost layer where the temperature gradient is larger (more positive)
than -2 K/km provided that the average lapse rate between this level and all higher levels within 2 km does not exceed 2 K/km
(World Meteorological Organization, 1992). It is often determined from data resampled on a grid corresponding to a wider
range of vertical spacings from below 1 km (e.g., significant pressure levels) and not always from the raw radiosonde data (see,
e.g., Reichler et al. 2003, and references therein).

From the radiosonde profiles the lapse rate tropopause altitudes were determined and served as our benchmark. Only in
those cases when the lapse rate tropopause determination failed, the cold point tropopause was used as benchmark instead. In
subsequent steps, the profiles were systematically degraded using averaging kernels of different shapes and altitude resolutions,
in order to investigate a possible vertical displacement of the apparent tropopause.

**Table 1.** Statistics for lapse rate tropopause displacements

**Nairobi**

| | |
|---|---|
| Latitude | 1.3°S |
| Longitude | 36.8°E |
| Sample Size | 452 |
| Years | 2007-1018 |

| Altitude Resolution (km) | Mean Deviation (km) | Standard Deviation (km) | Minimum Deviation (km) | Median Deviation (km) | Maximum Deviation (km) |
|---|---|---|---|---|---|
| 1.0 | 0.22 | 0.83 | -2.19 | 0.10 | 5.32 |
| 2.0 | -0.00 | 0.52 | -2.19 | 0.02 | 3.11 |
| 3.0 | -0.13 | 0.41 | -2.51 | -0.07 | 0.73 |
| 4.0 | -0.18 | 0.42 | -2.50 | -0.14 | 0.75 |
| 5.0 | -0.22 | 0.42 | -2.44 | -0.20 | 0.85 |

**Hilo**

| | |
|---|---|
| Latitude | 19.4°N |
| Longitude | 155.4°W |
| Sample Size | 540 |
| Years | 2007-2018 |

| Altitude Resolution (km) | Mean Deviation (km) | Standard Deviation (km) | Minimum Deviation (km) | Median Deviation (km) | Maximum Deviation (km) |
|---|---|---|---|---|---|
| 1.0 | -0.24 | 0.73 | -3.27 | -0.07 | 2.90 |
| 2.0 | -0.34 | 0.74 | -3.31 | -0.15 | 1.04 |
| 3.0 | -0.40 | 0.76 | -3.46 | -0.23 | 1.50 |
| 4.0 | -0.44 | 0.77 | -3.43 | -0.28 | 1.31 |
| 5.0 | -0.46 | 0.75 | -3.37 | -0.34 | 1.25 |

**Munich**

| | |
|---|---|
| Latitude | 47.8°N |
| Longitude | 10.9°E |
| Sample Size | 297 |
| Years | 2007-2018 |

| Altitude Resolution (km) | Mean Deviation (km) | Standard Deviation (km) | Minimum Deviation (km) | Median Deviation (km) | Maximum Deviation (km) |
|---|---|---|---|---|---|
| 1.0 | -0.59 | 0.98 | -4.81 | -0.36 | 3.00 |
| 2.0 | -0.69 | 1.02 | -6.46 | -0.38 | 2.83 |
| 3.0 | -0.73 | 1.05 | -6.43 | -0.39 | 0.27 |
| 4.0 | -0.77 | 1.20 | -6.38 | -0.37 | 0.37 |
| 5.0 | -0.76 | 1.25 | -6.29 | -0.33 | 0.57 |

**Greifswald**

| | |
|---|---|
| Latitude | 54.1°N |
| Longitude | 13.4°E |
| Sample Size | 275 |
| Years | 2007-2018 |

| Altitude Resolution (km) | Mean Deviation (km) | Standard Deviation (km) | Minimum Deviation (km) | Median Deviation (km) | Maximum Deviation (km) |
|---|---|---|---|---|---|
| 1.0 | -0.31 | 0.85 | -4.83 | -0.25 | 4.54 |
| 2.0 | -0.46 | 0.78 | -5.40 | -0.27 | 2.64 |
| 3.0 | -0.59 | 0.84 | -6.21 | -0.27 | 0.20 |
| 4.0 | -0.59 | 0.91 | -6.19 | -0.25 | 0.24 |
| 5.0 | -0.45 | 0.90 | -6.13 | -0.21 | 0.33 |

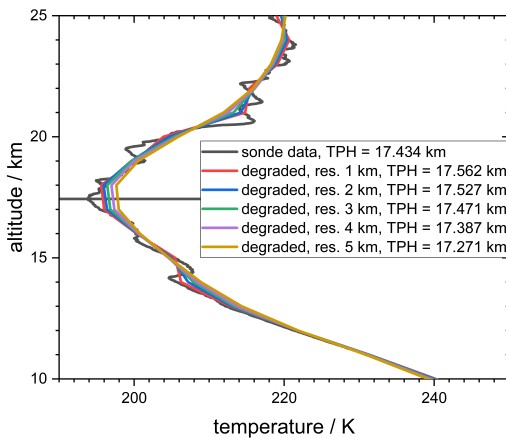

**Figure 1.** A radiosonde temperature profile measured at Nairobi, 1.3°S 36.8°E on 18 August 2010, 7:35 UT, along with a set of degraded profiles (Gaussian kernel) with resolutions from 1 to 5 km. The horizontal dark line indicates the original tropopause altitude. The inferred tropopause heights (TPH) are reported in the legend.

### 3.1.1 Gaussian averaging kernels

To test the dependence of apparent tropopause altitudes on the vertical resolution of the temperature profile, we use Gaussian-shaped averaging kernels to smooth the original radiosonde profiles. For this purpose we use the radiosonde data on their native grid. The smoothed profiles are sampled on a 1-km altitude grid. Vertical resolutions in terms of full widths at half maximum of 1 to 5 km were tested. Since we are not interested in the contribution by any a priori profile but only in the degradation of the vertical profiles by a degraded altitude resolution, we use Eq. (9), with the necessary modification to cope with the irregular input grid.

An example of an original radiosonde profile and a set of smoothed profiles are shown in Figure 1. These smoothed profiles were used to determine the tropopause altitudes according to the lapse rate criterion. Histograms of resulting vertical tropopause displacements for all profiles listed in Table 1 are shown in Figure 2. They are all well-behaved in a sense that they have no pronounced secondary modes. The histograms indicate an underlying left-skewed distribution. These asymmetries are attributed to the shape of the temperature profiles itself. A less resolved profile typically has a less negative lapse rate at lower altitudes than the better resolved profile. The tropopause determination scheme procedes from bottom upwards and the threshold will thus first be met already at lower altitudes.

The average tropopause displacements as a function of vertical resolution are reported in Table 1 and shown in Fig. 3. Tropopause altitudes inferred from coarser reduced data tend to be lower than those determined from the original sonde data. For a resolution of 3 km the mean (median) tropopause altitude displacement was found to be -130 to -730 m (-70 to -390 m). In most cases larger displacements were found for the midlatitudinal than for the tropical data, and the median is less affected than

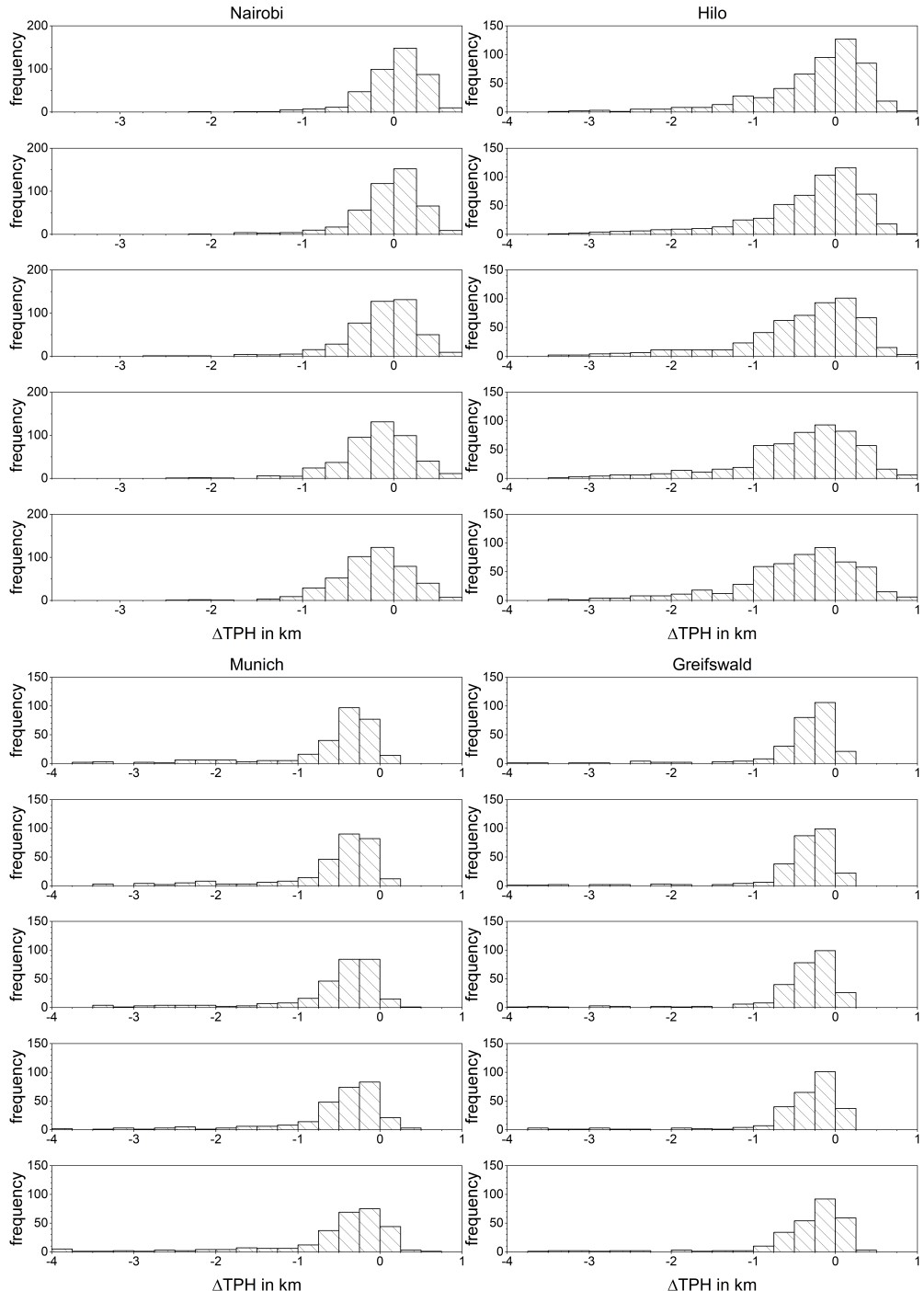

**Figure 2.** Histograms of mean tropopause height (ΔTPH) offsets (ΔTPH) of degraded temperature profiles (Gaussian kernel) with resolutions of 1 to 5 km in steps of 1 km (from top to bottom) for Nairobi (top left), Hilo (top right), Munich (bottom left) and Greifswald (bottom right).

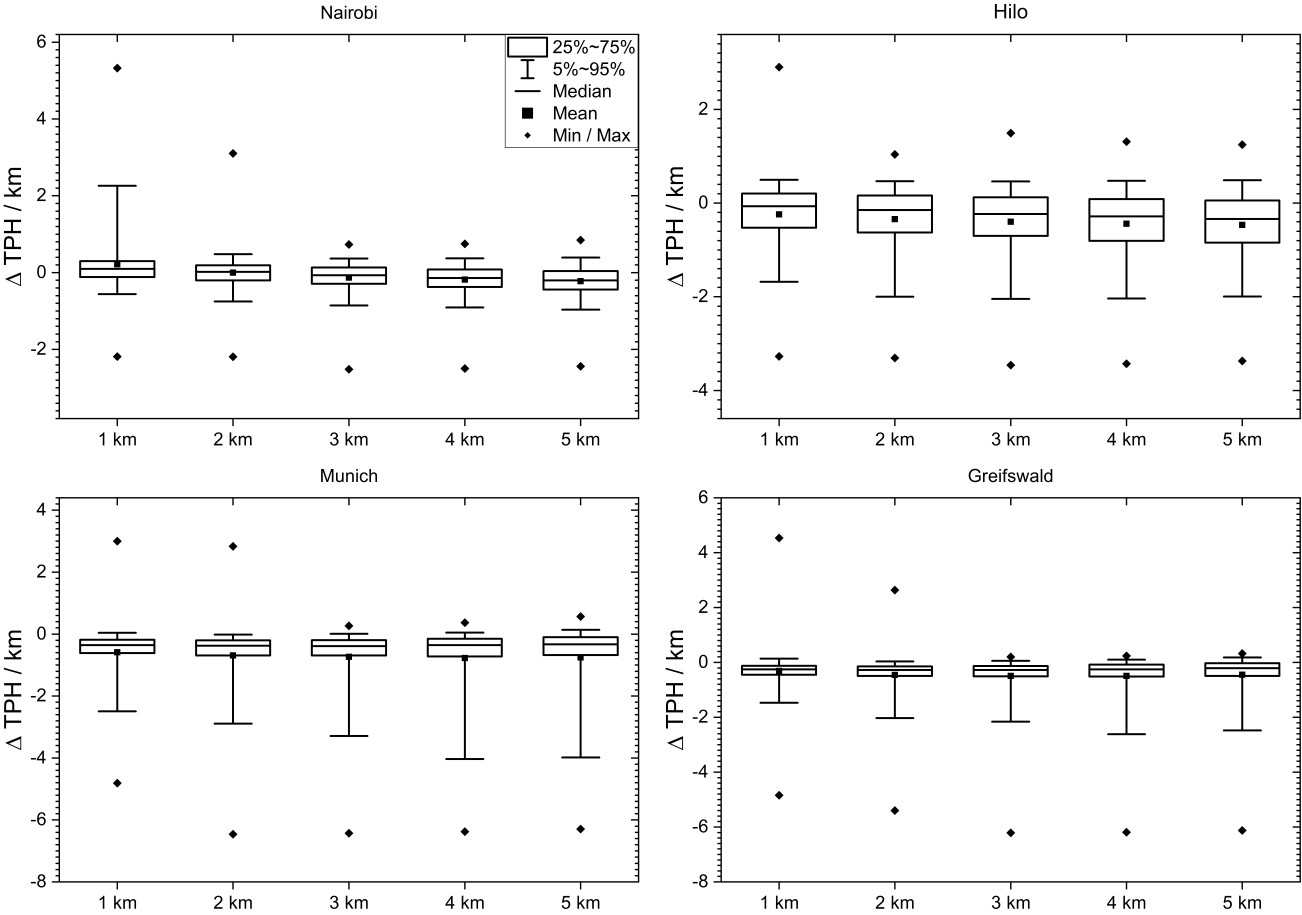

**Figure 3.** The lapse rate tropopause displacement for four geolocations as a function of vertical resolution for Gaussian kernels. Boxes represent both the second and the third quartile. The "error bars" represent the central 90% quantile.

the mean at all resolutions. The 5 to 95 percentile range increases for coarser resolutions and reaches saturation at a resolution beyond 3 km for the midlatitudinal stations. For Nairobi it covers a particularly wide range of displacements for 1 km vertical resolution.

### 3.1.2 MIPAS averaging kernels

5   To complement the analysis based on idealized averaging kernels, two exemplary case studies have been performed using averaging kernels characterizing temperature retrievals from Michelson Interferometer for Passive Atmospheric Sounding (MIPAS, Fischer et al. 2008) measurements. MIPAS was an infrared limb emission spectrometer operating on the Envisat research satellite. One of its data products was global temperature distributions from the upper troposphere to the mesosphere (von Clarmann et al., 2003b, 2009). We complement the theoretical study presented above with an assessment of how MIPAS temperature av-

**Table 2.** Nairobi MIPAS collocations

| Date | 3 Aug 2010 | 23 Sep 2010 |
|---|---|---|
| Difference Time (min.) | 70 | 35 |
| Difference Latitude ($^\circ$) | -2.8 | -0.1 |
| Difference Longitude ($^\circ$) | 0.0 | 1.3 |
| Tropopause Displacement (km) | -0.35 | -1.08 |

eraging kernels affect the lapse rate tropopause determination. For this case study we use averaging kernels and the a priori of two MIPAS retrievals which are spatiotemporally as close as possible to the radiosonde measurements (Table 2).

Since MIPAS averaging kernels are provided on a 1-km altitude grid, we use the radiosonde profiles resampled on a 1-km vertical grid, using Equation (6) with a $\mathbf{V}$ matrix for linear interpolation. Since the MIPAS averaging kernels ($\mathbf{A}_{\mathrm{MIPAS}}$) are

routinely produced on a 1-km grid (see, Fig. 4 for an example), they can then be conveniently applied to these resampled radiosonde profiles. The application of the averaging kernels as a filter function given by

$$\boldsymbol{x}_{\mathrm{degraded}} = \mathbf{A}_{\mathrm{MIPAS}}\boldsymbol{x}_{\mathrm{radiosonde}} \tag{11}$$

yields the radiosonde profile at the vertical resolution of the MIPAS temperatures. It does, however, not include the contribution of the a priori profiles used in the MIPAS retrieval. A more realistic transformation, which provides the radiosonde profiles as

MIPAS would see them, involves Eq. (3). Its application to the problem under investigation reads

$$\begin{aligned} \boldsymbol{x}_{\mathrm{degraded}} \quad = \quad & \mathbf{A}_{\mathrm{MIPAS}}\boldsymbol{x}_{\mathrm{radiosonde}} \\ & +(\mathbf{I} - \mathbf{A}_{\mathrm{MIPAS}})\boldsymbol{x}_{\mathrm{ERA-Interim}}, \end{aligned} \tag{12}$$

where $\boldsymbol{x}_{\mathrm{ERA-Interim}}$ are temperature profiles extracted from ECMWF ERA-Interim analyses (Dee et al., 2011), which were used as a priori information for the MIPAS retrievals. Since actual MIPAS measurement data are not used directly but only for

the calculation of the averaging kernels, and since the goal is to isolate the effect of the averaging kernel, the noise term is not considered here.

The resulting profile $\boldsymbol{x}_{\mathrm{degraded}}$ is the radiosonde profile as MIPAS would have seen it, if it had made a noise-free measurement exactly at the place and time of the radiosonde measurements. Again, the effect of the reduced resolution along with the effect of the a priori temperature profiles, on the tropopause altitude determination is investigated. Resulting displacements are

shown in the last row of Table 2. In the August case the displacement of -0.35 km lies within the standard deviation obtained for a 3-km Gaussian kernel, while in the September case it is considerably larger, with -1.08 km. The sign of the displacements found in the MIPAS case studies agrees with that found for the application of the Gaussian kernel.

The consideration of the prior information used is important. The reason is roughly this. In remote sensing applications, instead of referring to the resulting profile, the altitude resolution refers to the difference between the resulting profile and the

a priori profile. The fine structure of the prior information is propagated into the resulting profile retrieval and only corrections on a larger scale originate from the MIPAS measurements. We do not expect any systematicity with respect to this effect and

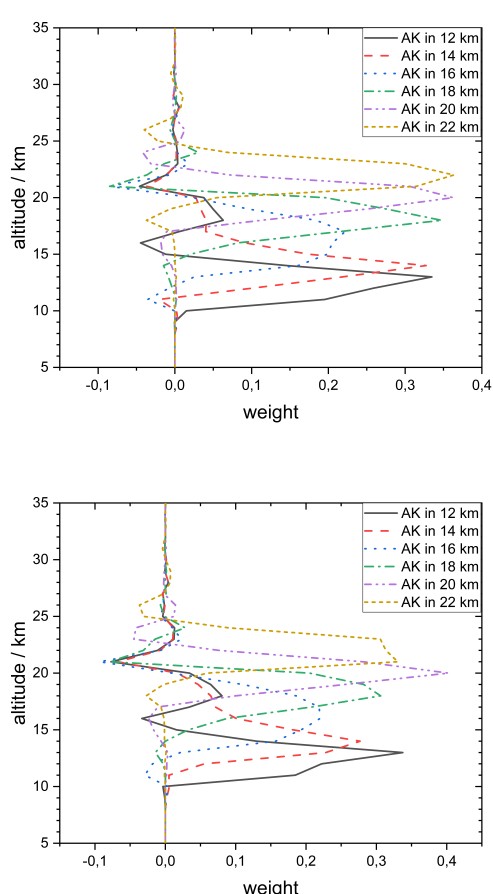

**Figure 4.** Averaging kernels of a MIPAS measurements near Nairobi on 3 Aug 2010 (top) and 23 Sep 2010 (bottom). The MIPAS altitude resolution is altitude dependent and varies between 2.7 and 4.5 km below 22 km altitude, with typical values of around 3.0 km. To avoid overly busy plots, only every other averaging kernel is shown.

we see no way to predict whether the smoothing effect or the fine structure of the a priori profile dominates the tropopause displacement.

For the comparison of the results obtained with the MIPAS averaging kernels and those obtained with the Gaussian averaging kernels, a caveat is adequate. Although coincidences under assessment are quite close, any spatiotemporal mismatch between the MIPAS and the sonde data can contribute to the displacement. Only MIPAS averaging kernels and a priori information are used for the comparison but no MIPAS data. The a priori data is in this case the ERA-Interim temperature profile, which depends on the MIPAS geolocation and measurement time. Any tropopause altitude difference between the ERA-Interim temperature profile at the MIPAS time and measurement location and at the radiosonde time and measurement location will map onto the degraded profile and adds a further component of uncertainty which can hardly be distinguished from the tropopause altitude offset caused by the degraded vertical resolution.

The problem discussed in this Section, that the use of a structured a priori profiles adds additional complication to the assessment of the tropopause displacement obviously is of no concern when a pure smoothing constraint in combination with a flat a priori profile is used for the retrieval.

## 3.2 The Cold Point Tropopause

In addition to the sensitivity of the lapse rate tropopause altitude to the vertical resolution of the temperature profile, also cold point tropopause altitudes were investigated. Again Gaussian averaging kernels were assessed (Fig. 5).

Here, the degrading with the Gaussian kernel was performed directly on the radiosonde profiles on their original grids and the cold point tropopauses were determined. Histograms of related tangent altitude displacements for resolutions from 1 to 5 km are shown in Fig. 6, while the dependence of the cold point tropopause altitude on the vertical resolution is shown in Fig. 7.

As for the lapse rate tropopause, also the cold point tropopause altitudes inferred from coarser resolved temperature profiles are lower than those inferred from the original profiles (Table 3). For the tropical stations and resolutions under investigation, the mean and median tropopause altitude displacement exceeds that of the lapse rate tropopause sizeably. The only exception is the vertical resolution of 1 km. Here the cold point tropopause appears to be less sensitive to the degraded resolution than the lapse rate tropopause. For the midlatitudinal stations Munich and Greifswald the cold point tropopause altitude is less sensitive to coarser resolutions than the lapse rate tropopause altitude in terms of the mean displacement. For the median displacent no clear superiority of one of the tropopause definitions could be identified.

MIPAS atmospheric state data as retrieved with the processor operated at the Institute of Meteorology and Climate Research (IMK) in cooperation with the Instituto de Astrofísica de Andalucía (IAA) (von Clarmann et al., 2009) are represented on a 1-km grid, and data of satellite instruments of similar vertical resolution are typically sampled on even coarser grids. Determination of the cold point tropopause on such a grid would thus by far be dominated by sampling effects. Tropopause shifts of a magnitude as determined with the Gaussian kernel thus cannot be safely resolved.

**Table 3.** Statistics for cold point tropopause displacements

| Nairobi | |
|---|---|
| Latitude | 1.3°S |
| Longitude | 36.8°E |
| Sample Size | 452 |
| Years | 2007-2018 |

| Altitude Resolution (km) | Mean Deviation (km) | Standard Deviation (km) | Minimum Deviation (km) | Median Deviation (km) | Maximum Deviation (km) |
|---|---|---|---|---|---|
| 1.0 | -0.16 | 0.13 | -1.30 | -0.15 | 0.43 |
| 2.0 | -0.35 | 0.22 | -1.65 | -0.31 | 0.09 |
| 3.0 | -0.51 | 0.34 | -2.68 | -0.46 | 0.59 |
| 4.0 | -0.66 | 0.43 | -3.20 | -0.60 | 0.51 |
| 5.0 | -0.75 | 0.47 | -3.54 | -0.69 | 0.29 |

| Hilo | |
|---|---|
| Latitude | 19.4°N |
| Longitude | 155.4°W |
| Sample Size | 540 |
| Years | 2007-2018 |

| Altitude Resolution (km) | Mean Deviation (km) | Standard Deviation (km) | Minimum Deviation (km) | Median Deviation (km) | Maximum Deviation (km) |
|---|---|---|---|---|---|
| 1.0 | -0.18 | 0.31 | -2.38 | -0.16 | 2.05 |
| 2.0 | -0.42 | 0.46 | -3.16 | -0.35 | 2.19 |
| 3.0 | -0.61 | 0.55 | -3.02 | -0.51 | 1.61 |
| 4.0 | -0.74 | 0.71 | -4.72 | -0.62 | 4.37 |
| 5.0 | -0.87 | 0.77 | -5.03 | -0.71 | 2.47 |

| Munich | |
|---|---|
| Latitude | 47.8°N |
| Longitude | 10.9°E |
| Sample Size | 297 |
| Years | 2007-2018 |

| Altitude Resolution (km) | Mean Deviation (km) | Standard Deviation (km) | Minimum Deviation (km) | Median Deviation (km) | Maximum Deviation (km) |
|---|---|---|---|---|---|
| 1.0 | -0.15 | 0.18 | -0.86 | -0.14 | 2.24 |
| 2.0 | -0.35 | 0.32 | -3.40 | -0.30 | 1.59 |
| 3.0 | -0.53 | 0.56 | -5.29 | -0.37 | 0.67 |
| 4.0 | -0.64 | 0.73 | -4.80 | -0.42 | 0.38 |
| 5.0 | -0.71 | 0.97 | -6.57 | -0.39 | 2.01 |

| Greifswald | |
|---|---|
| Latitude | 54.1°N |
| Longitude | 13.4°E |
| Sample Size | 275 |
| Years | 2007-2018 |

| Altitude Resolution (km) | Mean Deviation (km) | Standard Deviation (km) | Minimum Deviation (km) | Median Deviation (km) | Maximum Deviation (km) |
|---|---|---|---|---|---|
| 1.0 | -0.14 | 0.10 | -0.63 | -0.14 | 0.10 |
| 2.0 | -0.28 | 0.26 | -0.25 | -0.23 | 0.20 |
| 3.0 | -0.39 | 0.47 | -4.76 | -0.28 | 0.21 |
| 4.0 | -0.45 | 0.65 | -6.95 | -0.28 | 0.22 |
| 5.0 | -0.46 | 0.77 | -6.52 | -0.25 | 0.98 |

**Table 4.** Statistics for cold point water vapour saturation mixing ratio errors

| Nairobi | |
|---|---|
| Latitude | 1.3°S |
| Longitude | 36.8°E |
| Sample Size | 452 |
| Years | 2007-2018 |

| Altitude Resolution (km) | Mean Deviation (ppmv) | Standard Deviation (ppmv) | Minimum Deviation (ppmv) | Median Deviation (ppmv) | Maximum Deviation (ppmv) |
|---|---|---|---|---|---|
| 1.0 | 1.10 | 0.70 | -0.12 | 0.93 | 4.53 |
| 2.0 | 2.20 | 1.19 | 0.14 | 1.92 | 8.16 |
| 3.0 | 3.31 | 1.59 | 0.72 | 2.93 | 12.1 |
| 4.0 | 4.48 | 1.92 | 1.55 | 4.04 | 15.5 |
| 5.0 | 5.75 | 2.25 | 2.39 | 5.22 | 18.7 |

| Hilo | |
|---|---|
| Latitude | 19.4°N |
| Longitude | 155.4°W |
| Sample Size | 540 |
| Years | 2007-2018 |

| Altitude Resolution (km) | Mean Deviation (ppmv) | Standard Deviation (ppmv) | Minimum Deviation (ppmv) | Median Deviation (ppmv) | Maximum Deviation (ppmv) |
|---|---|---|---|---|---|
| 1.0 | 2.37 | 2.41 | -3.76 | 1.88 | 28.8 |
| 2.0 | 3.80 | 2.78 | -2.69 | 3.19 | 29.3 |
| 3.0 | 5.10 | 3.09 | -1.83 | 4.39 | 29.4 |
| 4.0 | 6.50 | 3.43 | -1.31 | 5.71 | 29.4 |
| 5.0 | 8.01 | 3.75 | -0.60 | 7.14 | 31.2 |

| Munich | |
|---|---|
| Latitude | 47.8°N |
| Longitude | 10.9°E |
| Sample Size | 297 |
| Years | 2007-2018 |

| Altitude Resolution (km) | Mean Deviation (ppmv) | Standard Deviation (ppmv) | Minimum Deviation (ppmv) | Median Deviation (ppmv) | Maximum Deviation (ppmv) |
|---|---|---|---|---|---|
| 1.0 | 7.03 | 4.69 | 0.33 | 5.72 | 37.3 |
| 2.0 | 13.1 | 7.35 | 3.53 | 11.4 | 51.0 |
| 3.0 | 18.8 | 9.65 | 5.04 | 16.7 | 64.5 |
| 4.0 | 24.5 | 12.1 | 7.20 | 21.9 | 81.8 |
| 5.0 | 30.6 | 14.7 | 7.98 | 27.7 | 99.0 |

| Greifswald | |
|---|---|
| Latitude | 54.1°N |
| Longitude | 13.4°E |
| Sample Size | 275 |
| Years | 2007-2018 |

| Altitude Resolution (km) | Mean Deviation (ppmv) | Standard Deviation (ppmv) | Minimum Deviation (ppmv) | Median Deviation (ppmv) | Maximum Deviation (ppmv) |
|---|---|---|---|---|---|
| 1.0 | 7.74 | 4.89 | 1.03 | 6.54 | 34.2 |
| 2.0 | 15.2 | 8.36 | 2.78 | 13.5 | 58.6 |
| 3.0 | 22.9 | 12.1 | 4.96 | 19.8 | 83.6 |
| 4.0 | 31.0 | 16.2 | 7.38 | 27.4 | 110. |
| 5.0 | 39.5 | 20.3 | 9.84 | 35.3 | 136. |

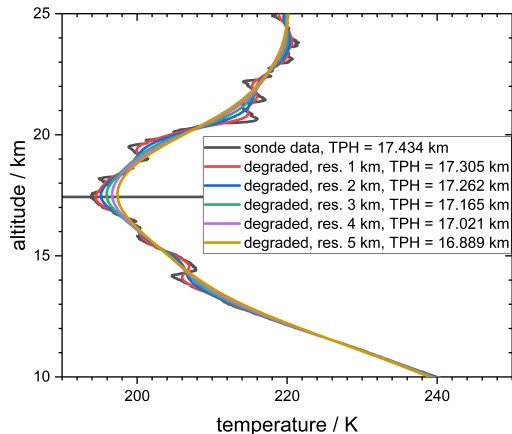

**Figure 5.** An example of a radiosonde profile, Nairobi, 1.27°S 36.8°E, 7 July 2010, at original vertical resolution along with degraded profiles at vertical resolutions of 1 to 5 km. Contrary to the example shown in Fig. 1, the smoothed profile is represented on the native radiosonde grid, not on the 1-km regular grid. For details, see Fig. 1.

## 4 Implications for Water Vapour Content

The cold point temperature largely determines the water vapour content of air entering the stratosphere. Thus, temperature profiles with finite vertical resolution affect the estimated saturation mixing ration of water vapour. For the tropical stations the related error in the saturation mixing ration seems to be fairly proportional to the vertical resolution. Reasonably good agreement between the mean and the median errors is found (Tab. 4 and Fig. 8). In the tropics, which are the entry region of air into the stratosphere, the error in the saturation mixing ratio in units of ppmv is about one or two times the resolution in km. For the midlatitudinal stations the error is much larger, with approximately seven ppmv per km of vertical resolution. These large errors, however, are of little concern in a global context because the midlatitudinal tropopause is not the preferred pathway of tropospheric air into the stratosphere. Admittedly, on smaller scales other transport pathways may be relevant (Anderson et al., 2012). The tropical maritime station Hilo stands out in a sense that the range of differences between saturation mixing ratios inferred from the original temperature profile and those inferred from degraded temperature profiles is large even for temperature profiles of 1 km vertical resolution and does not show a clear dependence on vertical resolution between 1 km and 5 km.

## 5 Feasibility of Correction Schemes

Since degraded, i.e., less resolved, temperature profiles are thought to exhibit less steep gradients, it suggests itself to adjust the lapse rate threshold in the WMO definition of the tropopause to compensate the smoothing effect. Doing this, one might

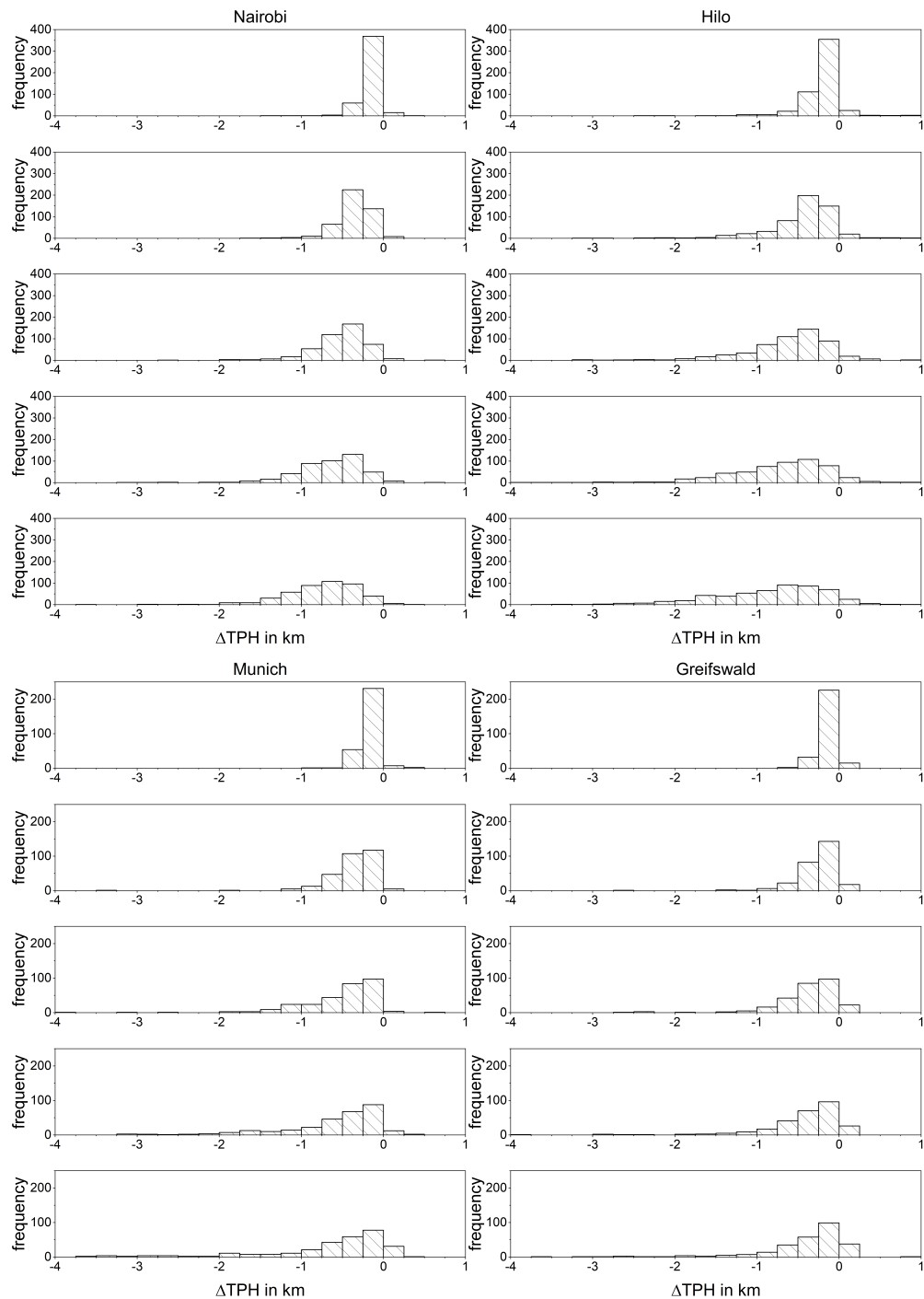

**Figure 6.** Histograms of tropopause height (TPH) offsets of degraded temperature profiles (Gaussian kernel) with resolutions of 1 to 5 km in steps of 1 km (from top to bottom) for Nairobi (top left), Hilo (top right), Munich (bottom left) and Greifswald (bottom right).

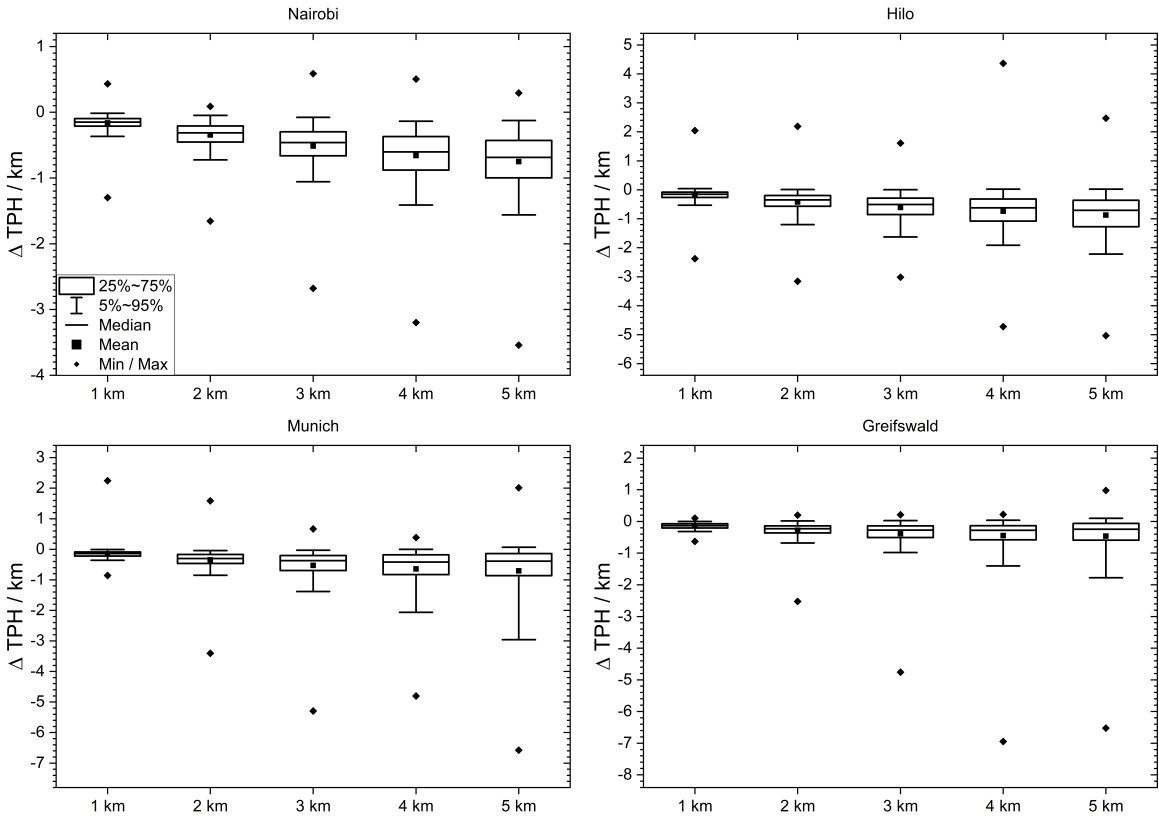

**Figure 7.** The cold point tropopause displacements for four geolocations as a function of the vertical resolution for Gaussian kernels. For details, see Fig. 3.

expect to find the tropopause at the correct altitude even from degraded profiles. Obviously, the threshold, if any useful, must be a function of the vertical resolution of the temperature profile used.

We have searched for a temperature gradient that, when applied to the coarse resolution profiles, reproduces the same tropopause altitude as the -2 K/km gradient applied to the original profiles. Fig. 9 shows histograms of the obtained adjusted lapse rate criteria for vertical resolutions of 1 to 5 km for all four stations under investigation. There is a tendency that lapse rates between -1 and -2 K/km are more adequate for applications to coarsely resolved profiles.

For the tropical stations Nairobi and Hilo, there is a clear tendency that for coarser vertical resolutions a smaller absolute value of the lapse rate would be more adequate to determine the tropopause. The spread, however, is very large (Table 5). The standard deviations of the optimal lapse rates (1.28–1.84 K/km) are approximately the same as the absolute values of the optimal lapses rates themselve (1.26–2.28 K/km), and even exceed them in some cases (Nairobi for 4 and 5 km resolution and Hilo for 5 km resolution). Often the standard deviation even exceeds the difference of between the nominal lapse rate of 2 K/km and the optimal lapse rate by a factor of ten or more. E.g. this difference is 0.11 K/km for Nairobi profiles at 2 km

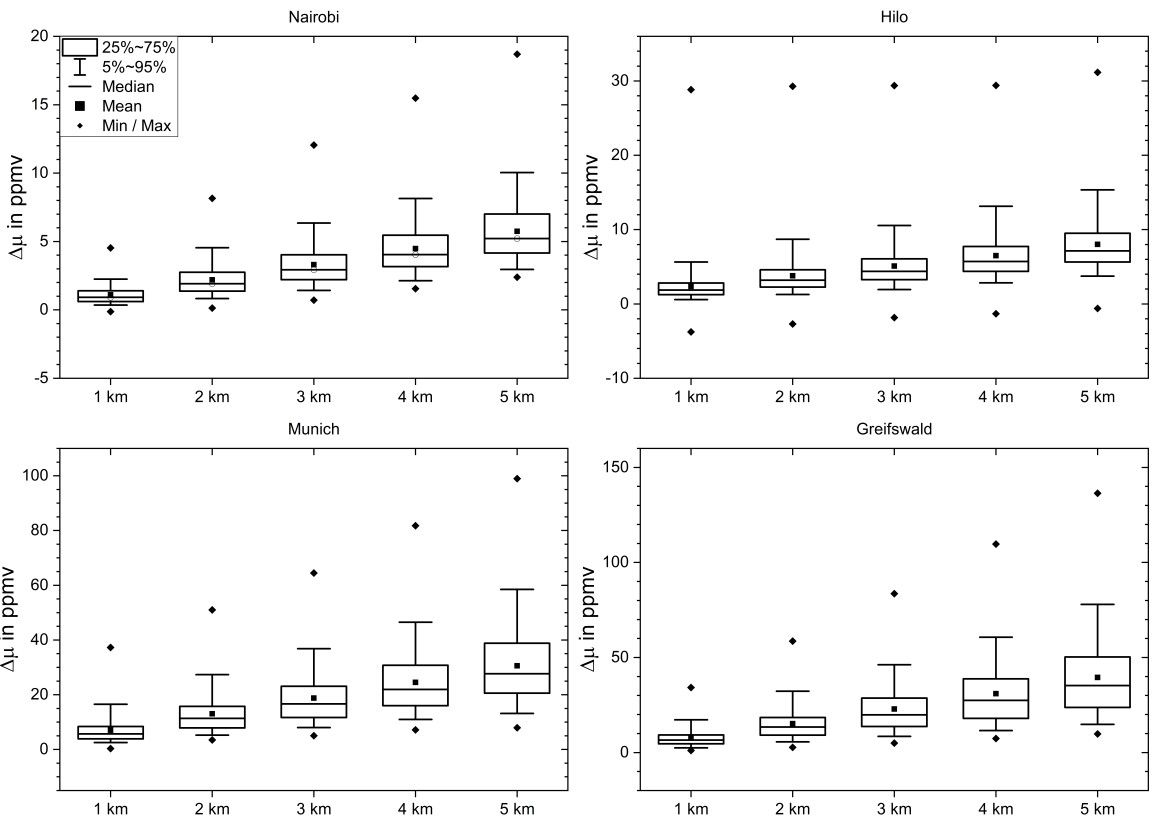

**Figure 8.** The effect of the vertical resolution and the related displacement of the cold point tropopause altitude on the water vapour saturation mixing ratio for four geolocations as a function of the vertical resolution for Gaussian kernels. For details, see Fig. 3.

vertical resolution while the standard deviation is 1.63 K/km. Thus we cannot recommend to use adjusted lapse rates to infer tropical tropopause altitudes from coarsely resolved temperature profiles.

The situation is different for midlatitudinal temperature profiles. The optimal lapse rate criterion turns out to be -1.3 K/km (Table 5). This value was found to be representative for both Munich and Greifswald. For Munich profiles the spread (0.62–
5  0.85 K/km) is little more than half of the absolute mean optimal value, and for Greifswald it is even less than a third (0.39–0.41 K/km). For Munich the spread is of similar size as the difference between the nominal and the mean optimal lapse rates, and for Greifswald it is only little larger than half this difference. Astonishingly enough the lapse rate was found to have a very weak dependence on the resolution on the vertical resolution, suggesting that the optimal lapse rate is not a continuous function of the vertical resolution but that this is a kind of threshold problem where the discontinuity is located at vertical resolutions
10  even better than 1 km. We are confident that for midlatitudinal temperature profiles the inductive inference of adapted lapse rate criteria can indeed improve tropopause altitude determination from coarsely resolved profiles.

**Table 5.** Statistics for lapse rate adjustments

|  | Nairobi | | | | |
|---|---|---|---|---|---|
| Latitude | 1.3°S | | | | |
| Longitude | 36.8°E | | | | |
| Sample Size | 452 | | | | |
| Years | 2007-2018 | | | | |
| Altitude Resolution (km) | Mean optimal lapse rate (K/km) | Standard Deviation (K/km) | Minimum (K/km) | Median (K/km) | Maximum (K/km) |
| 1.0 | -2.28 | 1.84 | -7.30 | -2.46 | 2.35 |
| 2.0 | -1.89 | 1.63 | -5.89 | -2.03 | 2.70 |
| 3.0 | -1.55 | 1.51 | -4.97 | -1.66 | 2.57 |
| 4.0 | -1.35 | 1.41 | -4.85 | -1.41 | 2.49 |
| 5.0 | -1.26 | 1.32 | -4.76 | -1.31 | 2.51 |

|  | Hilo | | | | |
|---|---|---|---|---|---|
| Latitude | 19.4°N | | | | |
| Longitude | 155.4°W | | | | |
| Sample Size | 540 | | | | |
| Years | 2007-2018 | | | | |
| Altitude Resolution (km) | Mean optimal lapse rate (K/km) | Standard Deviation (K/km) | Minimum (K/km) | Median (K/km) | Maximum (K/km) |
| 1.0 | -1.93 | 1.62 | -7.02 | -1.73 | 3.15 |
| 2.0 | -1.70 | 1.46 | -6.39 | -1.55 | 2.25 |
| 3.0 | -1.50 | 1.38 | -5.90 | -1.38 | 2.09 |
| 4.0 | -1.36 | 1.33 | -5.57 | -1.22 | 2.11 |
| 5.0 | -1.27 | 1.28 | -5.31 | -1.15 | 2.07 |

|  | Munich | | | | |
|---|---|---|---|---|---|
| Latitude | 47.8°N | | | | |
| Longitude | 10.9°E | | | | |
| Sample Size | 297 | | | | |
| Years | 2007-2018 | | | | |
| Altitude Resolution (km) | Mean optimal lapse rate (K/km) | Standard Deviation (K/km) | Minimum (K/km) | Median (K/km) | Maximum (K/km) |
| 1.0 | -1.30 | 0.85 | -2.78 | -1.52 | 2.34 |
| 2.0 | -1.29 | 0.78 | -2.71 | -1.49 | 2.19 |
| 3.0 | -1.30 | 0.70 | -2.67 | -1.47 | 1.40 |
| 4.0 | -1.30 | 0.65 | -2.64 | -1.45 | 1.15 |
| 5.0 | -1.29 | 0.62 | -2.60 | -1.42 | 0.97 |

|  | Greifswald | | | | |
|---|---|---|---|---|---|
| Latitude | 54.1°N | | | | |
| Longitude | 13.4°E | | | | |
| Sample Size | 275 | | | | |
| Years | 2007-2018 | | | | |
| Altitude Resolution (km) | Mean optimal lapse rate (K/km) | Standard Deviation (K/km) | Minimum (K/km) | Median (K/km) | Maximum (K/km) |
| 1.0 | -1.34 | 0.41 | -2.91 | -1.30 | 0.34 |
| 2.0 | -1.32 | 0.39 | -2.84 | -1.29 | 0.00 |
| 3.0 | -1.30 | 0.39 | -2.78 | -1.28 | 0.00 |
| 4.0 | -1.29 | 0.39 | -2.75 | -1.26 | 0.00 |
| 5.0 | -1.29 | 0.40 | -2.79 | -1.25 | 0.00 |

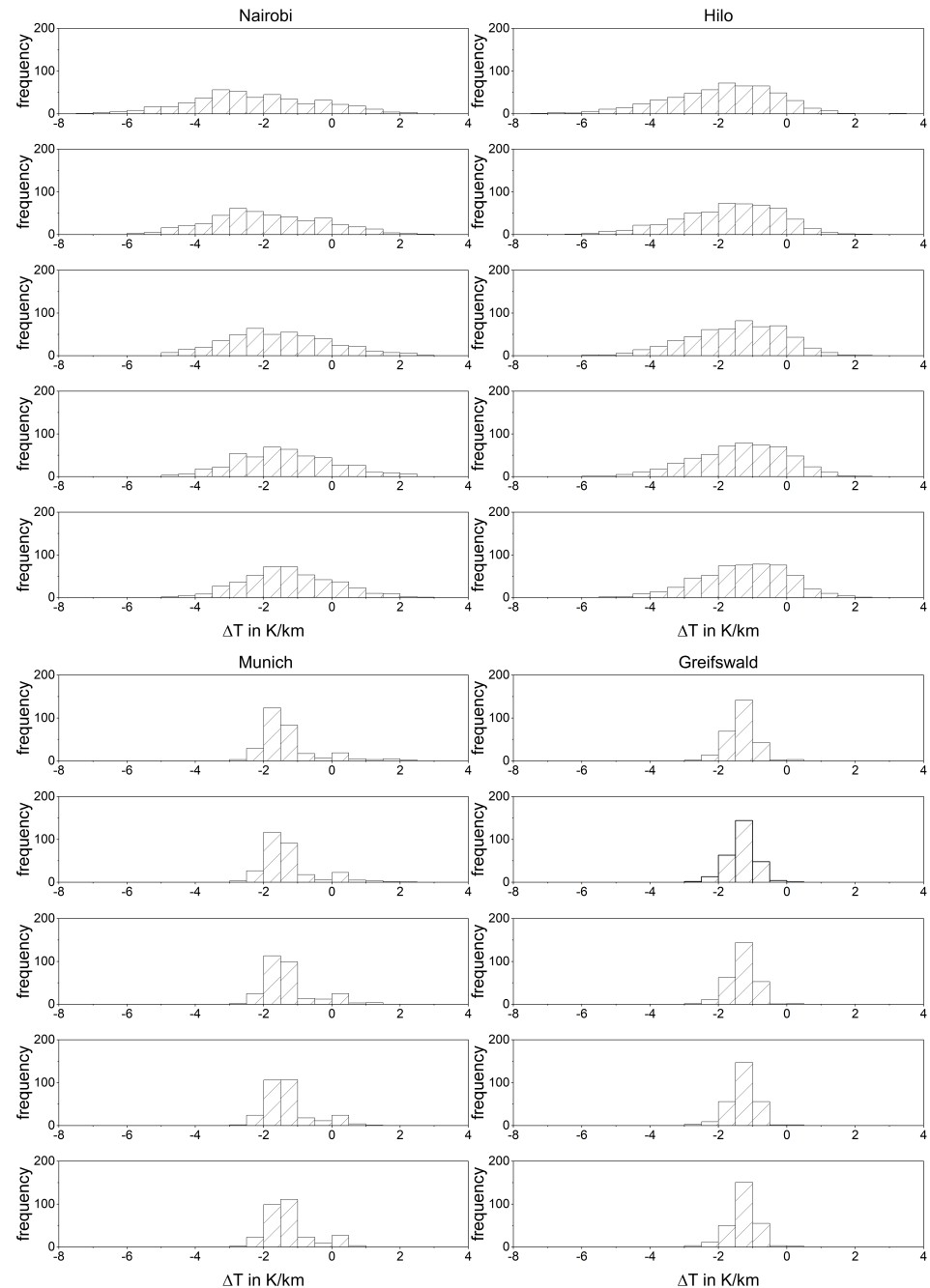

**Figure 9.** Histograms of optimal lapse rates for tropopause determination from coarsely resolved temperature profiles for four stations. Vertical resolutions vary from 1 km (top) to 5 km (bottom).

# 6 Discussion and Conclusion

In the tropical region the determination of both, the lapse rate and the cold point tropopause altitude from temperature profiles of degraded altitude resolution, typically leads to an underestimation of the tropopause height. The mean magnitudes of this effect range from 0 to about 500 m for altitude resolutions of 1 to 5 km. Often considerably larger effects are found for the cold point tropopause. For midlatitudinal temperature profiles larger tropopause altitude displacements were found, and, broadly speaking, the cold point tropopause turns out to be less sensitive to vertical resolution issues. This suggests that, while in the tropics the cold point tropopause is commonly used, in the case of coarsely resolved profiles, the lapse rate tropopause appears to be more robust, and *vice versa* for midlatitudinal atmospheres. However, adaptive, resolution dependent, lapse rates can improve the tropopause determination in mid-latitudes. In contrast, the variability of the tropopause dislocation is fairly large in the tropics such that the recommendation of an inductively generalized correction scheme for tropical tropopause heights seems audacious and even inappropriate to us.

Given the variety of retrieval schemes used to infer temperature profiles from satellite measurements, the following caveats with respects to the generality of our results need to be discussed.

Often satellite retrievals use a retrieval scheme similar to Eq. (1) along with a highly structured a priori profile $x_a$. A retrieval with a vertical resolution that is significantly coarser than that of $x_a$ will modify only the coarse structure of the temperature profile, while the fine structure of $x_a$ will survive the retrieval process. This is because the resolution of the retrieval as defined by the averaging kernel refers, rigorously speaking, not to the resulting profile, but to the difference between the resulting profile and the a priori profile. Related effects on the tropopause displacement are then complicated to predict, because it depends largely on the surviving fine structure. Tropopause determination procedures do not distinguish between the a priori contribution and the measurement contribution to the final temperature profile. The large displacements along with the large scatter found in the analysis of MIPAS averaging kernels (Section 3.1.2) are attributed to this effect. In consequence, it seems to be more appropriate to use smooth a priori profiles if retrieved temperature profiles are intended to be used for tropopause altitude determination.

Another issue of concern is the retrieval grid of the temperature retrieval. It is the exception rather than the rule that limb sounders use a retrieval grid which is about a factor of three finer than the vertical resolution, as the MIPAS dataset used here. More often the vertical grid is close to the vertical resolution of the retrieval. In these cases, the tropopause altitude determination is limited directly by the sampling of the retrieval and not by its resolution.

Our results may have implications for other than remotely sensed temperature profiles of limited vertical resolutions, such as model or analysis data. While our methods seem appropriate also for related investigations, such problems are beyond the scope of this paper.

*Author contributions.* The paper originated from NK's BSc project supervised by PB and TvC. NK coded the software, performed the case studies, generated the figures and contributed to the writing of the paper. PB wrote part of the introduction and formulated the BSc project

with TvC. TvC suggested the study, coordinated the writing of the paper, and contributed major parts of the text. All authors discussed the results, suggested conclusions, and contributed to the final wording.

*Data availability.* Tropical Radiosonde data were obtained from the SHADOZ website: https://tropo.gsfc.nasa.gov/shadoz/. German ra-
diosonde data were obtained from German Weather Servive via ftp://ftp-cdc.dwd.de/. MIPAS data are available via https://www.imk-
asf.kit.edu/english/308.php.

*Competing interests.* TvC is associate editor of AMT but has not been involved in the evaluation of this paper.

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
