# Peer review of "Tropopause altitude determination from temperature profiles of reduced altitude resolution"

_Atmospheric Measurement Techniques, 2018_

## Referee Comment (RC1) · Anonymous Referee #1 · 8 Feb 2019

The paper by Koenig et al, addresses an important topic concerning the determination of the tropopause altitude from high resolution data using low resolution kernels. These have to be applied e.g. when comparing high resolution vertical data to satellite observations, when filtering oder interpolating data sets of different vertical resolution. The authors provide a systematic analysis using SHADOZ sonde data. They test for the effect of different resolutions (kernels) on the WMO lapse rate tropopause and cold point tropopause altitudes. They find, that the tropopause altitudes of the kernel weighted fine scale profiles differ significantly from the tropopause altitudes of the original data. Importantly they conclude that there is no correction scheme applicable to account for the displacement.

The paper is sound and I regard it as highly relevant and recommend it for publication

almost as it is. The authors could easily expand the scientific importance of their work by extending their analysis adding a systematic analysis of the cold point temperatures as well for different resolutions. Since the temperatures are crucial for the analysis of humidity transport, this would add another important aspect to this nice and important study.

Specific: It would be interesting to plot the cold point temperature in the same way as Fig.7. Which implications does this have for the water vapor saturation mixing ratio?

Why are the differences mostly negative with decreasing resolution (Fig3, especially Fig.7)?

Technical: Are there differences in the kernels used in Fig.1 and Fig.6? If not, is Fig.6 necessary for the paper? p. 11, line 6. "coaErse"

---

## Referee Comment (RC2) · Anonymous Referee #2 · 25 Mar 2019

**General remarks:**

The manuscript by König et al. presents interesting aspects for the tropopause determination from temperatures profiles of restricted vertical resolution. This topic is not only crucial for satellite based temperature sounding but also for frequent analyses and applications of meteorological data sets. The mathematical background based on Rodgers (2000) is described in detail and allows a nearly correct an elegant description of the problem. In addition, it allows an accurate description and quantification of resulting error sources. The paper is well organized and written, and the scientific and technical objectives will fit to the scope of AMT. However, I have strong concerns that the paper in the current form is adequately addressing the scientific and technical standards of AMT, I am generally missing a more in depth

analysis and larger statistics (e.g. a larger set of radiosonde stations) to draw robust and meaningful conclusions. I can only recommend the publication of the study of König et al. after some major revisions and improvements. More detailed suggestions for improvements and comments are specified in the following sections.

**Major comments:**

The title of the paper promises more than the analysis and the final results can deliver. The tropopause (TP) determination of reduced altitude profiles - like announced - in only analyzed for one very specific instrument (MIPAS), but is a quite general problem and especially important for many studies taken into account meteorological data sets like ERA-Interim or MERRA. Very similar problems occur for these type of data, if you like to quantify the error in the TP determination for the relatively coarse vertical sampling around the TP compared the typical fine resolution of the radiosonde data. TP heights are not part of the meteorological data sets. It would be by far more interesting to apply the methodology in a more general approach , for example to the problem outlined above. Take these criticisms into account the author should change the title accordingly.

The abstract is extremely short and includes even repetitions ('3 km vertical resolution'). The reader may ask, are there as little results? This is also true for the conclusions and unfortunately my final impression of the presented study, although there seems a high potential in the formalism.

The data base and the statistical analyses have a couple of limitations, which need improvements in a new version of the paper:

a) The number of 30 radiosonde profile for only one station (Nairobi) is far too small for significant conclusions based on the presented analyses and statistics.

b) The selection of only one station representative for the tropics seems also critical. Are there any references for this simplification? I guess continental and coastal area can have quite different temperature profile (wave activity), also regions with strong and low convection activity. If the study likes to stay with its focus on the tropics, then more stations and coincidences with MIPAS should be taken into account.

c) More stations and profiles (by taken a longer time period) would also help to bypass the very coarse coincidence criteria applied in this study. A $\pm$ 1000 km in longitudinal direction and $\pm$500 km in latitude is by far too coarse to define a proper coincidence. In addition, I am missing a miss-time criterion in the manuscript?

d) All statistic plots suffer on the general problem of the study of rather low number of profiles/coincidences. For me it makes no sense to fit Gauss distributions to histograms or to present box-whisker plots for such low ensemble numbers.

Overall, I would recommend to apply the methodology not only to the tropics, because TP determination it is a general problem at all latitudes, which would give the study a much broader scientific relevance. In addition, the authors should think about to apply the formalism to temperature profiles of meteorological analyses, which would give a much broader scientific community a tool or reference to quantify uncertainties in the tropopause determination (e.g. for tropopause related coordinates, definition of the transport barrier).

**Minor comments:**

The authors should reference in the introduction to other limb based remote sensing analyses in former publications or to more general publications highlighting the difficulties and importance of an accurate tropopause determination (e.g. Pan and Munchak, 2011, Peevy et al., 2012, or Spang et al., 2015).

What is the effect of the higher resolved and vertically more structured *a priori* profiles (e.g. ERA interim) on the MIPAS temperature retrievals and finally on the MIPAS TP determination. Can you quantify this effect with your methodology?

Why do the authors include Figures 6 with no additional information compared to Figure 1, what's new or has to be highlighted here? Tropopause heights (radiosonde and potentially for different degraded resolutions) should be superimposed in both Figures.

Section 4 on the feasibility of correction schemes is missing a detailed analysis and the description is too short. This section has currently not the substance for a full section in a paper, it's just a result for a paragraph. Again the number of profiles is not sufficient. I am wondering why the author made the analysis with such a limited data set of radiosonde profiles and MIPAS profiles. It will be easy - but of coarse additional work - to extend the complete study to a larger database and to draw better and more profound conclusions.

**References:**

Pan, L. L., and L. A. Munchak: Relationship of cloud top to the tropopause and jet structure from CALIPSO data, J. Geophys. Res., 116, D12201, doi:10.1029/2010JD015462, 2011.

Peevey, T. R., J. C. Gille, C. E. Randall, and A. Kunz: Investigation of double tropopause spatial and temporal global variability utilizing High Resolution Dynamics Limb Sounder temperature observations, J. Geophys. Res., 117, D01105, doi:10.1029/2011JD016443, 2012.

Rodgers, C. D.: Inverse Methods for Atmospheric Sounding: Theory and Practice, vol.

2 of Series on Atmospheric, Oceanic and Planetary Physics, F. W. Taylor, ed., World Scientific, Singapore, New Jersey, London, Hong Kong, 2000.

Spang, R., Günther, G., Riese, M., Hoffmann, L., Müller, R., and Griessbach, S.: Satellite observations of cirrus clouds in the Northern Hemisphere lowermost stratosphere, Atmos. Chem. Phys., 15, 927-950, https://doi.org/10.5194/acp-15-927-2015, 2015.

---

## Author Comment (AC1) · 29 Apr 2019

**Comment:** *The paper by Koenig et al, addresses an important topic concerning the determination of the tropopause altitude from high resolution data using low resolution kernels. These have to be applied e.g. when comparing high resolution vertical data to satellite observations, when filtering oder interpolating data sets of different vertical resolution. The authors provide a systematic analysis using SHADOZ sonde data. They test for the effect of different resolutions (kernels) on the WMO lapse rate tropopause and cold point tropopause altitudes. They find, that the tropopause altitudes of the kernel weighted fine scale profiles differ significantly from the tropopause altitudes of the original data. Importantly they conclude that there is no correction scheme applicable to account for the displacement.*

[Figure]

*The paper is sound and I regard it as highly relevant and recommend it for publication almost as it is. The authors could easily expand the scientific importance of their work by extending their analysis adding a systematic analysis of the cold point temperatures as well for different resolutions. Since the temperatures are crucial for the analysis of humidity transport, this would add another important aspect to this nice and important study.*

**Reply:** Excellent idea! This will be included in the revised version.

**Comment:** *Specific: It would be interesting to plot the cold point temperature in the same way as Fig. 7. Which implications does this have for the water vapor saturation mixing ratio?*

**Reply:** Plots of related cold point temperature will be shown. Implications for the water vapor saturation mixing ratio will be discussed.

**Comment:** *Why are the differences mostly negative with decreasing resolution (Fig 3, especially Fig. 7)?*

**Reply:** This is due to nonlinear shapes of the temperature profile, i.e., the profile is not fully described by the gradients above and below the tropopause.

**Comment:** *Technical: Are there differences in the kernels used in Fig.1 and Fig.6? If not, is Fig.6 necessary for the paper?*

**Reply:** After inclusion of additional results the selection of figures will have to be

re-decided anyway.

**Comment:** *p. 11, line 6. "coaErse"*

**Reply:** Thanks for spotting.
* * *

---

## Author Comment (AC2) · 29 Apr 2019

**Comment: General remarks:**

The manuscript by König et al. presents interesting aspects for the tropopause determination from temperatures profiles of restricted vertical resolution. This topic is not only crucial for satellite based temperature sounding but also for frequent analyses and applications of meteorological data sets. The mathematical background based on Rodgers (2000) is described in detail and allows a nearly correct and elegant description of the problem. In addition, it allows an accurate description and quantification of resulting error sources. The paper is well organized and written, and the scientific and technical objectives will fit to the scope of AMT.

Reply: We thank the reviewer for this encouraging evaluation.

**Comment:** However, I have strong concerns that the paper in the current form is adequately addressing the scientific and technical standards of AMT, I am generally missing a more in depth analysis and larger statistics (e.g. a larger set of radiosonde stations) to draw robust and meaningful conclusions. I can only recommend the publication of the study of König et al. after some major revisions and improvements. More detailed suggestions for improvements and comments are specified in the following sections.

**Reply:** Please see our comments below.

**Comment:** Major comments:**

The title of the paper promises more than the analysis and the final results can deliver.

**Reply:** We do not quite see the point. The title is "Tropopause altitude determination from temperature profiles of reduced altitude resolution" and this is exactly what we critically assess in the paper. As stated below, we will replace the term 'profiles' by 'measurements'.

**Comment:** The tropopause (TP) determination of reduced altitude profiles - like announced - is only analyzed for one very specific instrument (MIPAS)[...]

**Reply:** This is not true. We investigate into this effect for a series of idealized instruments with altitude resolutions of 1 to 5 km. This analysis is applicable to a wide range of instruments. Analyses based on averaging kernels from real instruments suffer from
the fact that the averaging kernels have to be taken as they are, and the dependence of the effect on variation of the altitude resolution thus cannot be assessed. Thus the focus of the study is on the idealized averaging kernels where the altitude resolution can be varied. MIPAS results are presented in addition as an illustrative case study.

**Comment:** [...] but is a quite general problem and especially important for many studies taken into account meteorological data sets like ERA-Interim or MERRA. Very similar problems occur for these type of data, if you like to quantify the error in the TP determination for the relatively coarse vertical sampling around the TP compared the typical fine resolution of the radiosonde data. TP heights are not part of the meteorological data sets. It would be by far more interesting to apply the methodology in a more general approach, for example to the problem outlined above.

**Reply:** This study was performed with application to satellite measurements in mind. The applicability of the methodology presented is an added value but for a paper in Atmospheric Measurement Techniques we find it adequate to restrict the study to atmospheric measurements.

**Comment:** Take these criticisms into account the author should change the title accordingly.

**Reply:** To avoid wrong expectations we change the title to "Tropopause altitude determination from temperature profile measurements of reduced altitude resolution."

**Comment:** The abstract is extremely short and includes even repetitions ('3 km vertical resolution'). The reader may ask, are there as little results? This is also true for the conclusions and unfortunately my final impression of the presented study, although

AMTD
there seems a high potential in the formalism.

**Reply:** The additional analysis performed in reply to both reviews will add length to the abstract and the short sections.

**Comment:** The data base and the statistical analyses have a couple of limitations, which need improvements in a new version of the paper: a) The number of 30 radiosonde profile for only one station (Nairobi) is far too small for significant conclusions based on the presented analyses and statistics.

**Reply:** We wonder in which sense the reviewer uses the term 'significant' here. The context of sample size suggests that it is meant as a statistical technical term, while the context of 'conclusion' suggests that the term is used in a colloquial sense. In statistics only differences can be significant. In our case, the differences are considerably larger than the uncertainties of even a single profile. Thus we can say even for a single profile that the effect is significant. For any given profile, the effect is deterministic. Note that we do not recommend any inductive correction based on the mean tropopause altitude displacement. In this case significance would be defined by the standard error of the mean displacement and the displacement. But, as said above, we neither propose nor endorse such an inductive correction. Nevertheless we have increased the sample size considerably to gain a better idea on the representativeness of our results.

**Comment:** b) The selection of only one station representative for the tropics seems also critical. Are there any references for this simplification? I guess continental and coastal area can have quite different temperature profile (wave activity), also regions with strong and low convection activity. If the study likes to stay with its focus on the tropics, then more stations and coincidences with MIPAS should be taken into account.
**Reply:** We do not see how a statistic should become more robust by making the sample more inhomogeneous. Nevertheless we take this suggestion and include analyses of other sites but we do not think that it would be a good idea to merge all this into one statistic.

**Comment:** c) More stations and profiles (by taken a longer time period) would also help to bypass the very coarse coincidence criteria applied in this study.  $A \pm 1000$  km in longitudinal direction and  $\pm 500$  km in latitude is by far too coarse to define a proper coincidence. In addition, I am missing a miss-time criterion in the manuscript?

**Reply:** This reads as if the reviewer has misunderstood our approach. Coincidence criteria would indeed be far too coarse if we had compared MIPAS profiles to radiosonde profiles but we did not do that. We only have applied MIPAS averaging kernels to the sonde profiles and compare the original sonde profiles to those where the averaging kernel has been applied. MIPAS averaging kernels are only weekly or at best moderately temperature-dependent, and thus this issue is, in this application, a higher order effect.

**Comment:** *d)* All statistic plots suffer on the general problem of the study of rather low number of profiles/coincidences. For me it makes no sense to fit Gauss distributions to histograms or to present box-whisker plots for such low ensemble numbers.

**Reply:** The sample size will be increased. The Gaussian curves were only illustrative, that is to say, no conclusions have beed drawn from them; thus they can easily be removed. All statistics comes directly from the data without any detour over Gaussian pdfs.
**Comment:** Overall, I would recommend to apply the methodology not only to the tropics, because TP determination it is a general problem at all latitudes, which would give the study a much broader scientific relevance.

Reply: Agreed, other sites will be included.

**Comment:** In addition, the authors should think about to apply the formalism to temperature profiles of meteorological analyses, which would give a much broader scientific community a tool or reference to quantify uncertainties in the tropopause determination (e.g. for tropopause related coordinates, definition of the transport barrier).

**Reply:** We are glad if the theoretical part helps also for these applications but we think that for a paper in a journal on measurement techniques it is justified to restrict the analysis to measurements.

**Comment: Minor comments:**

The authors should reference in the introduction to other limb based remote sensing analyses in former publications or to more general publications highlighting the difficulties and importance of an accurate tropopause determination (e.g. Pan and Munchak, 2011, Peevy et al., 2012, or Spang et al., 2015).

**Reply: Agreed.**

**Comment:** What is the effect of the higher resolved and vertically more structured a priori profiles (e.g. ERA interim) on the MIPAS temperature retrievals and finally on the
MIPAS TP determination. Can you quantify this effect with your methodology?

**Reply:** This depends on the relation between the amplitudes of the fine structure and those resolved by the instrument. No general statement can be made on this.

**Comment:** Why do the authors include Figures 6 with no additional information compared to Figure 1, what is new or has to be highlighted here? Tropopause heights (radiosonde and potentially for different degraded resolutions) should be superimposed in both Figures.

**Reply:** With additional data available the decision how to best present the results will have to be drawn from the scratch anyway.

**Comment:** Section 4 on the feasibility of correction schemes is missing a detailed analysis and the description is too short. This section has currently not the substance for a full section in a paper, it's just a result for a paragraph. Again the number of profiles is not sufficient. I am wondering why the author made the analysis with such a limited data set of radiosonde profiles and MIPAS profiles. It will be easy - but of coarse additional work - to extend the complete study to a larger database and to draw better and more profound conclusions.

**Reply:** More data will be included. Again, we suspect that there is a misunderstanding because no MIPAS profiles have been used, only MIPAS averaging kernels. The bulk of the study relies on idealized averaging kernels corresponding to a range of altitude resolutions.

---

## Author Response (AR1)

**Reviewer 1**

**Comment:** *The paper by Koenig et al, addresses an important topic concerning the determination of the tropopause altitude from high resolution data using low resolution kernels. These have to be applied e.g. when comparing high resolution vertical data to satellite observations, when filtering oder interpolating data sets of different vertical resolution. The authors provide a systematic analysis using SHADOZ sonde data. They test for the effect of different resolutions (kernels) on the WMO lapse rate tropopause and cold point tropopause altitudes. They find, that the tropopause altitudes of the kernel weighted fine scale profiles differ significantly from the tropopause altitudes of the original data. Importantly they conclude that there is no correction scheme applicable to account for the displacement.*
*The paper is sound and I regard it as highly relevant and recommend it for publication almost as it is. The authors could easily expand the scientific importance of their work by extending their analysis adding a systematic analysis of the cold point temperatures as well for different resolutions. Since the temperatures are crucial for the analysis of humidity transport, this would add another important aspect to this nice and important study.*

**Reply:** Excellent idea!

**Action:** A new Section on water vapour saturation mixing ratios has been included, along with a table and a figure.

**Comment:** *Specific: It would be interesting to plot the cold point temperature in the same way as Fig. 7. Which implications does this have for the water vapor saturation mixing ratio?*

**Reply:** Agreed.

**Action:** Plots and tables of related cold point temperature have been included. Implications for the water vapor saturation mixing ratio are now discussed.

**Comment:** *Why are the differences mostly negative with decreasing resolution (Fig. 3, especially Fig. 7)?*

**Reply:** For the lapse rate tropopause, this is because weaker gradients are found already at lower altitudes if the resolution is coarser. And the tropopause search goes bottom up. For the cold point tropopause, we have no general explanation. This depends on the particular wiggles of each individual temperature profile.

**Action:** An explanation has been added to the text.

**Comment:** *Technical: Are there differences in the kernels used in Fig.1 and Fig.6? If not, is Fig.6 necessary for the paper?*

**Reply:** The main difference is the sampling of the degraded profiles. It is the sampling of the radiosonde profiles in Fig 6 (smoother curves) while it is a 1 km grid in Fig 1. Please note that the figure numbering has changed, it is now Fig. 5.

**Action:** We have changed the line styles to make this difference better discernable.

**Comment:** *p. 11, line 6. "coaErse"*

**Reply:** Thanks for spotting.

**Action:** This typo has been corrected.

**Reviewer 2:**

**Comment:** *General remarks:*
*The manuscript by König et al. presents interesting aspects for the tropopause determination from temperatures profiles of restricted vertical resolution. This topic is not only crucial for satellite based temperature sounding but also for frequent analyses and applications of meteorological data sets. The mathematical background based on Rodgers (2000) is described in detail and allows a nearly correct and elegant description of the problem. In addition, it allows an accurate description and quantification of resulting error sources. The paper is well organized and written, and the scientific and technical objectives will fit to the scope of AMT.*

**Reply:** We thank the reviewer for this encouraging evaluation.

**Action:** none

**Comment:** *However, I have strong concerns that the paper in the current form is adequately addressing the scientific and technical standards of AMT, I am generally missing a more in depth analysis and larger statistics (e.g. a larger set of radiosonde stations) to draw robust and meaningful conclusions. I can only recommend the publication of the study of König et al. after some major revisions and improvements. More detailed suggestions for improvements and comments are specified in the following sections.*

**Reply:** Please see our answers below.

**Action:** Those of the answers below.

**Comment:** *Major comments:*
*The title of the paper promises more than the analysis and the final results can*

*deliver.*

**Reply:** We do not quite see the point. The title is "Tropopause altitude determination from temperature profiles of reduced altitude resolution" and this is exactly what we critically assess in the paper.

**Action:** As stated below, we have reworded the title: "Tropopause altitude determination from temperature profile measurements of reduced vertical resolution". Since we have now found a correction scheme at least for midlatitudes, the reviewer might find the title more appropriate now.

**Comment:** *The tropopause (TP) determination of reduced altitude profiles - like announced - is only analyzed for one very specific instrument (MIPAS)[...]*

**Reply:** This is not true. We investigate into this effect for a series of idealized instruments with altitude resolutions of 1 to 5 km. This analysis is applicable to a wide range of instruments. Analyses based on averaging kernels from real instruments suffer from the fact that the averaging kernels have to be taken as they are, and the dependence of the effect on variation of the altitude resolution thus cannot be systematically assessed. Thus the focus of the study is on the idealized averaging kernels where the altitude resolution can be varied. MIPAS results are presented in addition as an illustrative case study.

**Action:** none

**Comment:** *[...] but is a quite general problem and especially important for many studies taken into account meteorological data sets like ERA-Interim or MERRA. Very similar problems occur for these type of data, if you like to quantify the error in the TP determination for the relatively coarse vertical sampling around the TP compared the typical fine resolution of the radiosonde data. TP heights are not part of the meteorological data sets. It would be by far more interesting to apply the methodology in a more general approach , for example to the problem outlined above. Take these criticisms into account the author should change the title accordingly.*

**Reply:** This study was performed with application to satellite measurements in mind. The applicability of the methodology presented is an added value but for a paper in Atmospheric Measurement Techniques we find it adequate to restrict the study to atmospheric measurements.

**Action:** To avoid wrong expectations we change the title to "Tropopause altitude determination from temperature profile measurements of reduced altitude resolution." We mention that the range of applicability of our concept may be wider but that this is beyond the scope of our paper.

**Comment:** *The abstract is extremely short and includes even repetitions ('3*

*km vertical resolution'). The reader may ask, are there as little results? This is also true for the conclusions and unfortunately my final impression of the presented study, although there seems a high potential in the formalism.*

**Reply:** The additional analysis performed in reply to both reviews added length to the abstract and the short sections.

**Action:** The additional results have been included in the abstract.

**Comment:** *The data base and the statistical analyses have a couple of limitations, which need improvements in a new version of the paper: a) The number of 30 radiosonde profile for only one station (Nairobi) is far too small for significant conclusions based on the presented analyses and statistics.*

**Reply:** We wonder in which sense the reviewer uses the term 'significant' here. The context of sample size suggests that it is meant as a statistical technical term, while the context of 'conclusion' suggests that the term is used in a colloquial sense. In statistics only differences can be significant. In our case, the differences are considerably larger than the uncertainties of even a single profile (which has nothing to do with the standard error of the mean or other related statistical quanties. It is the mere measurement error which is relevant here). Thus we can say even for a single profile that the effect is significant. For any given profile, the effect is deterministic. Note that in the original manuscript we did not recommend any inductive correction based on the mean tropopause altitude displacement on the basis of this limited data set.

**Action:** We have increased the sample size considerably to gain a better idea on the representativeness of our results.

**Comment:** *b) The selection of only one station representative for the tropics seems also critical. Are there any references for this simplification? I guess continental and coastal area can have quite different temperature profile (wave activity), also regions with strong and low convection activity. If the study likes to stay with its focus on the tropics, then more stations and coincidences with MIPAS should be taken into account.*

**Reply:** We do not see how a statistic should become more robust by making the sample more inhomogeneous. Nevertheless we take this suggestion and include analyses of other sites but we do not think that it would be a good idea to merge all this into one statistic.

**Action:** We have included one further tropical station and two stations in mid-latitudes.

**Comment:** *c) More stations and profiles (by taken a longer time period) would also help to bypass the very coarse coincidence criteria applied in this study. A*

± *1000 km in longitudinal direction and* ±*500 km in latitude is by far too coarse to define a proper coincidence. In addition, I am missing a miss-time criterion in the manuscript?*

**Reply:** This reads as if the reviewer has misunderstood our approach. Coincidence criteria would indeed be far too coarse if we had compared MIPAS profiles to radiosonde profiles but we did not do that. We only have applied MIPAS averaging kernels to the sonde profiles and compare the original sonde profiles to those where the averaging kernel has been applied. MIPAS averaging kernels are only weekly or at best moderately temperature-dependent, and thus this issue is, in this application, a higher order effect. The only problem left is geolocation-dependent priori information.

**Action:** We say now clearly in the text that application to MIPAS is meant only as an illustrative case study. We have reduced the number of MIPAS cases in order to guarantee fairly close collocations. We have removed all related statistics in order not to lead the reader astray towards misinterpretation of these results in a statistical sense.

**Comment:** *d) All statistic plots suffer on the general problem of the study of rather low number of profiles/coincidences. For me it makes no sense to fit Gauss distributions to histograms or to present box-whisker plots for such low ensemble numbers.*

**Reply:** Even in the original manuscript no conclusions were drawn from the fitted Gaussian distributions. All statistics came (and comes) directly from the data without any detour over Gaussian pdfs.

**Action:** Sample sizes have been increased. The Gaussian curves were removed.

**Comment:** *Overall, I would recommend to apply the methodology not only to the tropics, because TP determination it is a general problem at all latitudes, which would give the study a much broader scientific relevance.*

**Reply:** Agreed.

**Action:** Two sites in midlatitudes have been included.

**Comment:** *In addition, the authors should think about to apply the formalism to temperature profiles of meteorological analyses, which would give a much broader scientific community a tool or reference to quantify uncertainties in the tropopause determination (e.g. for tropopause related coordinates, definition of the transport barrier).*

**Reply:** We are glad if the theoretical part helps also for these applications but we think that for a paper in a journal on measurement techniques it is justified

to restrict the analysis to measurements.

**Action:** We mention the possible applicability of our concept to other types of data but state that this is beyond the scope of this paper.

**Comment:** *Minor comments:*
*The authors should reference in the introduction to other limb based remote sensing analyses in former publications or to more general publications highlighting the difficulties and importance of an accurate tropopause determination (e.g. Pan and Munchak, 2011, Peevy et al., 2012, or Spang et al., 2015).*

**Reply:** Agreed.

**Action:** These and some more references have been included.

**Comment:** *What is the effect of the higher resolved and vertically more structured a priori profiles (e.g. ERA interim) on the MIPAS temperature retrievals and finally on the MIPAS TP determination. Can you quantify this effect with your methodology?*

**Reply:** This depends on the relation between the amplitudes of the fine structure and those resolved by the instrument. No general statement can be made on this.

**Action:** Some discussion has been added.

**Comment:** *Why do the authors include Figures 6 with no additional information compared to Figure 1, what is new or has to be highlighted here? Tropopause heights (radiosonde and potentially for different degraded resolutions) should be superimposed in both Figures.*

**Reply:** The different characteristic (sampling of the smoothed profiles) was simply not visible with the line style chosen in the original manuscript.

**Action:** The line style of the smoothed profiles has been changed to solid lines to make the sampling characteristic better visible.

**Comment:** *Section 4 on the feasibility of correction schemes is missing a detailed analysis and the description is too short. This section has currently not the substance for a full section in a paper, it's just a result for a paragraph. Again the number of profiles is not sufficient. I am wondering why the author made the analysis with such a limited data set of radiosonde profiles and MIPAS profiles. It will be easy - but of coarse additional work - to extend the complete study to a larger database and to draw better and more profound conclusions.*

**Reply:** Again, we suspect that there is a misunderstanding because no MIPAS profiles have been used for this purpose. As expected, for Nairobi the scatter is, even after increasing the sampling size considerably, still way too large to recommend a correction. This does not come as a surprise, because the standard deviation does not converge towards zero for increasing sample size but converges towards the true standard deviation. Since the point of interest is the correction of individual profiles, it is the scatter which is relevant, not the standard error of the mean. The scatter of a sample with a size larger than approximately 30 commonly is typically fairly robust such that even the use of a student's t distribution instead of a Gaussian is considered unnecessary.

For the midlatitudinal stations we have found a fairly robust correction scheme which is now discussed in this section and adds length.

**Action:** Additional results have been included in this section.

**By the way:** This article was intended to be a "Technical Note". The manuscript was then published as a regular discussion paper because the format "Technical Note" is not supported by AMT. At the AMT editorial board meeting in Vienna 2019 one of us suggested to the AMT Executive Editors to introduce "Technical Note" as a new article format in AMT. Their reply was that this is not necessary because length is no criterion for AMT articles.

[revised manuscript text omitted]

---

## Referee Report (RR1)

**Review of the revised version of the AMT mansuscript:**

Tropopause altitude determination from temperature profile measurements of reduced vertical resolution

by: Nils König, Peter Braesicke, and Thomas von Clarmann

Overall the manuscript is substantially improved. The author did a good job by adding a couple of stations like suggested, which even show interesting differences to the mid-latitude tropopause, and by a more precise and detailed description of their results. In addition, the analysis of the water vapor entry value - suggested by the 2nd reviewer - is a value-added for the manuscript. I have only minor comments, questions, and suggestions. Consequently, this novel approach and analysis of limits in the tropopause determination should be ready for publication in AMT with only little effort for the authors.

**General comment:**

How do double or multiple tropopause events influence the results of the analysis? If two tropopauses are very close to each other (e.g. Mehta et al., 2011), then degraded vertical resolutions and TP criteria - like applied in the manuscript - may results in quite different results for the 1st tropopause. I would like to suggest, that the authors comment on the potential effects (e.g. biases) by double tropopause events. Maybe the positive outlier in Fig. 3 for Nairobi is affected by this effect.

**Specific comments:**

All line and page numbers are attributed to the track-changes version of the manuscript.

Abstract: Is it not a general rule for AMT to introduce acronyms already in the abstract (here: MIPAS)?

Abstract: The term 'tropopause displacement within each sample' might be a bit misleading. Do you mean 'each profile'?

Page 2, line 20: The Spang et al. (2015) is only analyzing ERA-i temperature profiles. The reference should move to the sentence before. The paper critically discussed, how accurate the 'true' tropopause height can be retrieved from the ERA-i temperature profiles.

p2,l23: 'assess' instead of 'asses' ?

p5, l27: The new sentence is confusing me a bit. The cold point tropopause is a separate analysis and there is no mixing of both analyses types, correct?

Fig. 1: Could you please add a horizontal line for the 'true' TPH, and for each degraded profile the numbers of the TPH into the legend box.

P7, l9: 'Figures 6', is the order of the figures correct? Fig.6 mentioned before Fig.3 is slightly confusing.

Fig. 4: Are the smaller peaks of the averaging kernels related to the tropopause location (~16 km)?

If yes, then please comment on this fact and it may be helpful to add the TP height to this figure as well.

p4, l7: 'MIPAS data are sampled on a 1-km grid' sounds like a sampling for MIPAS TP measurements of 1km. I think this is not correct. The minimal sampling is 1.5 km. I guess you mean something different. Please clarify.

**p14, l10 ff:**

'preferred pathway of tropospheric pathways into the stratosphere': the author should give a little more details on this topic (references and some text). For example, Anderson et al. (2012) postulated a severe imprint of deep convection on stratospheric water vapor and ozone in Summer over the USA.

Fig 8.: Any explanation why Nairobi looks so different to Hilo (drift to much larger dH2O/km with coarser resolution)?

**Technical comment:**

p22, l12: 'appropriate' sounds better to me than 'apt'

Fig. 3,7, and 8: Please, enlarge symbols for Mean and Min/Max, which are hard to spot on printout and screen.

**References:**

Metha, S. K., et al.: Multiple tropopauses in the tropics: A cold point approach, J. Geophys. Res., vol. 116, D20105, 2011.

Anderson, J. G., et al.: UV Dosage Levels in Summer: Increased Risk of Ozone Loss from Convectively Injected Water Vapor, DOI: 10.1126/science.1222978, Science 337, 835, 2012.

---

## Author Response (AR2)

**Authors' reply:**

The authors thank the reviewer for the helpful review.

**Comment:** *Review of the revised version of the AMT mansuscript:*
*Tropopause altitude determination from temperature profile measurements of reduced vertical resolution by: Nils Knig, Peter Braesicke, and Thomas von Clarmann*
*Overall the manuscript is substantially improved. The author did a good job by adding a couple of stations like suggested, which even show interesting differences to the mid-latitude tropopause, and by a more precise and detailed description of their results. In addition, the analysis of the water vapor entry value - suggested by the 2nd reviewer - is a value-added for the manuscript. I have only minor comments, questions, and suggestions. Consequently, this novel approach and analysis of limits in the tropopause determination should be ready for publication in AMT with only little effort for the authors.*

**Reply:** We thank the reviewer for the appreciation of our revision.

**Action:** None.

**Comment:** *General comment: How do double or multiple tropopause events influence the results of the analysis? If two tropopauses are very close to each other (e.g. Mehta et al., 2011), then degraded vertical resolutions and TP criteria - like applied in the manuscript - may results in quite different results for the 1st tropopause. I would like to suggest, that the authors comment on the potential effects (e.g. biases) by double tropopause events. Maybe the positive outlier in Fig. 3 for Nairobi is affected by this effect.*
*Reference: Metha, S. K., et al.: Multiple tropopauses in the tropics: A cold point approach, J. Geophys. Res., vol. 116, D20105, 2011.*

**Reply:** Our tropopause detection algorithm detects only the lowermost tropopause. We have checked the outlier; it cannot be attributed to a double tropopause. Since we have not systematically investigated double tropopauses, we are reluctant to include statement on this issue in the paper because these might be too speculative.

**Action:** None.

**Comment:** *Specific comments:*
*All line and page numbers are attributed to the track-changes version of the manuscript. Abstract: Is it not a general rule for AMT to introduce acronyms already in the abstract (here: MIPAS)?*

**Reply:** We agree.

**Action:** The acronym is now defined in the abstract.

**Comment:** *Abstract: The term 'tropopause displacement within each sample' might be a bit misleading. Do you mean 'each profile'?*

**Reply:** There is no spread of tropopause displacements within a single profile. However, we have reworded this for clarity.

**Action:** We now write "within each sample **of profiles**."

**Comment:** *p2,l23: 'assess' instead of 'asses' ?*

**Reply:** Thanks for spotting!

**Action:** Corrected.

**Comment:** *p5, l27: The new sentence is confusing me a bit. The cold point tropopause is a separate analysis and there is no mixing of both analyses types, correct?*

**Reply:** Yes, in principle theses are two separate analyses. However, there are some cases where, according to its definition, no lapse rate tropopause existed. In these exceptional cases the cold point tropopause was used as benchmark instead.

**Action:** We have added "...was used **as benchmark** instead."

**Comment:** *Fig. 1: Could you please add a horizontal line for the 'true' TPH, and for each degraded profile the numbers of the TPH into the legend box.*

**Reply:** Good idea.

**Action:** We have added the horizontal line which indicates the tropopause altitude and have included the numerical values of tropopause altitudes in the legend. The figure caption has been changed accordingly. Although not requested, we have applied the same changes to Fig. 5.

**Comment:** *P7, l9: 'Figures 6', is the order of the figures correct? Fig.6 mentioned before Fig.3 is slightly confusing.*

**Reply:** The order of figures was correct but there was an error in the respective labels. Thanks for spotting!

**Action:** The reference label of Fig 6 has been corrected.

**Comment:** *Fig. 4: Are the smaller peaks of the averaging kernels related to the tropopause location ( 16 km If yes, then please comment on this fact and it may be helpful to add the TP height to this figure as well.*

**Reply:** Among the authors there was no agreement what the reviewer means with "smaller peaks". Is the reviewer speaking about the (a) secondary peaks (side wiggles) or (b)does he refer to the fact the lower altitude averaging kernels have a slightly smaller peak and coarser resolution than those of higher altitudes?
(a) These side wiggles have nothing to do with the actual tropopause. Limb sounding averaging kernels represented on such fine grid have always such type of side wiggles. These are caused by the particular structure of the Jacobian.
(b) There are multiple causes. Of course cold parts of the atmosphere emit less radiance, which results in wider averaging kernels. But also the spectral analysis windows are chosen altitude-dependent, and saturation effects (non-linearity of radiative transfer) plays a role at these low altitudes. In one of the cases shown, the small and wide averaging kernels coincides with the tropopause, in the other it does not. Since there is no unambiguous relation, we prefer not to raise this issue in the paper.
**Action:** None

**Comment:** *p4, l7: 'MIPAS data are sampled on a 1-km grid' sounds like a sampling for MIPAS TP measurements of 1km. I think this is not correct. The minimal sampling is 1.5 km. I guess you mean something different. Please clarify.*

**Reply:** The reviewer is correct in that the tangent altitude spacing is coarser but we refer to the sampling of the atmospheric state variables, not the direct radiance measurements. Since the retrieval grid is in this case a 1-km grid, the atmospheric state variables are sampled on a 1-km grid. We concede that the term "sampling" is ambiguous and rewrite the text accordingly.

**Action:** We now write: "MIPAS atmospheric state data as retrieved with the processor operated at the Institute of Meteorology and Climate Research (IMK) in cooperation with the Instituto de Astrofísica de Andalucía (IAA) (von Clarmann et al., 2009) are represented on a 1-km grid...". The reference is necessary, because other MIPAS processors don't do it this way.

**Comment:** *p14, l10 ff: 'preferred pathway of tropospheric pathways into the stratosphere': the author should give a little more details on this topic (references and some text). For example, Anderson et al. (2012) postulated a severe imprint of deep convection on stratospheric water vapor and ozone in Summer over the USA.*
*Reference: Anderson, J. G., et al.: UV Dosage Levels in Summer: Increased Risk of Ozone Loss from Convectively Injected Water Vapor, DOI: 10.1126/science.1222978, Science 337, 835, 2012.*

**Reply:** We agree that ous statement was overgeneralizing.

**Action:** We now write: "These large errors, however, are of little concern in a global context because the midlatitudinal tropopause is not the preferred pathway of tropospheric air into the stratosphere. Admittedly, on smaller scales other transport pathways may be relevant (Anderson et al., 2012)"

**Comment:** *Fig 8.: Any explanation why Nairobi looks so different to Hilo (drift to much larger dH2O/km with coarser resolution)?*

**Reply:** Due to a technical mistake, in the last version Tables and Figure 8 were old versions. This, however, does not explain the effect under discussion. Hilo is the only tropical maritime station. We can only speculate that under such conditions the relationship between tempertures and water vapour content are different than at continental or midlatitudinal sites. We hestitate to offer an explanation because this would be speculative. However, we point at this distinctive feature in the paper.

**Action:** Tables 1, 3, 4 and 5 have been corrected and Fig. 8 has been replaced by the most recent version. Some corrected numbers required minor adjustments in the text. At the end of Section 4 we have added: "The tropical maritime station Hilo stands out in a sense that the range of differences between saturation mixing ratios inferred from the original temperature profile and those inferred from degraded temperature profiles is large even for temperature profiles of 1 km vertical resolution and does not show a clear dependence on vertical resolution between 1 km and 5 km".

**Comment:** *Technical comment: p22, l12: 'appropriate' sounds better to me than 'apt'*

**Reply:** Agreed.

**Action:** Changes as suggested.

**Comment:** *Fig. 3,7, and 8: Please, enlarge symbols for Mean and Min/Max, which are hard to spot on printout and screen.*

**Reply:** Agreed

**Action:** Done; another, better visible symbol has been chosen for the mean.

[revised manuscript text omitted]